# Hexameric and pentameric complexes of the ExbBD energizer in the Ton system

**Saori Maki-Yonekura[1], Rei Matsuoka[1], Yoshiki Yamashita[1], Hirofumi Shimizu[2], Maiko Tanaka[1], Fumie Iwabuki[1], Koji Yonekura[1]***

[1]Biostructural Mechanism Laboratory, RIKEN SPring-8 Center, Sayo, Japan; [2]Department of Molecular Physiology and Biophysics, Faculty of Medical Sciences, University of Fukui, Fukui, Japan

**Abstract** Gram-negative bacteria import essential nutrients such as iron and vitamin $B_{12}$ through outer membrane receptors. This process utilizes proton motive force harvested by the Ton system made up of three inner membrane proteins, ExbB, ExbD and TonB. ExbB and ExbD form the proton channel that energizes uptake through TonB. Recently, crystal structures suggest that the ExbB pentamer is the scaffold. Here, we present structures of hexameric complexes of ExbB and ExbD revealed by X-ray crystallography and single particle cryo-EM. Image analysis shows that hexameric and pentameric complexes coexist, with the proportion of hexamer increasing with pH. Channel current measurement and 2D crystallography support the existence and transition of the two oligomeric states in membranes. The hexameric complex consists of six ExbB subunits and three ExbD transmembrane helices enclosed within the central channel. We propose models for activation/inactivation associated with hexamer and pentamer formation and utilization of proton motive force.

DOI: https://doi.org/10.7554/eLife.35419.001

*For correspondence:
yone@spring8.or.jp

**Competing interests:** The authors declare that no competing interests exist.

## Introduction

Many essential biological processes are coupled to ion potentials across the lipid membrane. These include, amongst others, ATP synthesis (*Junge et al., 2009*), uptake of nutrients (*Faraldo-Gómez and Sansom, 2003*; *Noinaj et al., 2010*; *Krewulak and Vogel, 2011*; *Abramson et al., 2003*), protein secretion (*Tsukazaki et al., 2011*), multi-drug excretion (*Murakami et al., 2006*), and rotation of the flagellar motor (*Kojima, 2015*). The Ton complex (*Faraldo-Gómez and Sansom, 2003*; *Noinaj et al., 2010*; *Krewulak and Vogel, 2011*) in gram-negative bacteria is an interesting example of a molecular machine that depends on proton motive force (PMF). While other systems use this transmembrane (TM) energy at source (that is at the membrane with the proton gradient) the Ton system conveys it to the distant outer membrane, ~20 nm away from the energy source sited at the inner membrane (*Du et al., 2014*; *Matias et al., 2003*) (*Figure 1—figure supplement 1*). The Ton complex is composed of three proteins ExbB, ExbD and TonB located in the inner membrane. It utilizes PMF to take up diverse compounds such as iron-loaded siderophores, haem, and vitamin $B_{12}$ through outer membrane receptors (*Faraldo-Gómez and Sansom, 2003*; *Noinaj et al., 2010*; *Krewulak and Vogel, 2011*; *Schauer et al., 2008*). Toxins such as colicin and some antibiotics, as well as phages, hijack the Ton system to gain entry to these cells (*Noinaj et al., 2010*; *Krewulak and Vogel, 2011*; *Schauer et al., 2008*; *Lloubès et al., 2012*). It could perhaps be used, through drug targeting, to attack pathogenic gram-negative bacteria (*Neelapu et al., 2015*).

In response to proton flux through the Ton complex, the C-terminal domain of TonB interacts with TonB-dependent outer membrane receptors, which bind specific substrates. A conformational change releases the bound substrate from the receptor into the periplasmic space. Each substrate is then transported into the cell by its specific inner membrane transporters and related proteins

**eLife digest** Many biological processes that are essential for life are powered by the flow of ions across the membranes of cells. Similar to how energy is stored in the water behind a dam, energy is also stored when the concentration of ions on one side of a biological membrane is higher than it is on the other. When these ions then flow down this concentration gradient, the energy can be harnessed to power other processes.

In many bacteria, the concentration of hydrogen ions, or protons, is higher on the outside of the cell. When the protons flow down the concentration gradient, a protein complex called the Ton system in the bacteria's inner membrane harnesses the energy to transport various compounds, including essential nutrients, across the outer membrane, which is about 20 nanometres away. Toxins, and viruses that infect bacteria, can also hijack the Ton system to gain entry into these cells. This means that the Ton system could perhaps be targeted via drugs to treat bacterial infections.

Though the Ton system is important, structural information on this protein family is limited. The Ton complex is composed of three proteins – ExbB, ExbD and TonB – located in the bacteria's inner membrane. ExbB and ExbD together form a channel for the protons and the complex made from these two proteins can be thought of as the system's engine.

Maki-Yonekura et al. wanted to understand how the ExbB / ExbD complex works, which was challenging because the complex was not well suited to any single structural biology technique. To get around this issue, a combination of two techniques called X-ray crystallography and single particle cryo-EM were used. This approached revealed that the two proteins form complexes made up of either five or six ExbB subunits with one or three ExbD subunits, respectively. It also showed that the proteins transition between the two forms in a cell's membrane. More of the larger six-unit complex (also called a "hexamer") formed at higher pH. This is consistent with the increased flow of protons through the channel when the local conditions inside the cell become less acidic.

Based on these results, Maki-Yonekura et al. propose that some subunits in the core of the complex rotate to harness the energy from the flow of protons, and the number of subunits in the complex changes when it switches to become active or inactive. The discoveries may provide a new vision of dynamic membrane biology. Further studies are now needed to see how general this mechanism is in biology, and the new structural information could also be used to help develop more anti-bacterial drugs.

DOI: https://doi.org/10.7554/eLife.35419.002

(*Figure 1—figure supplement 1*). In all cases, the Ton complex initiates the process and works as the energizer for outer membrane substrate uptake. ExbB has three TM helices with a large cytoplasmic domain (*Celia et al., 2016*), and both ExbD and TonB are predicted to have single TM helices and a compact periplasmic domain (*Garcia-Herrero et al., 2007*; *Ködding et al., 2005*). ExbB and ExbD together form a proton channel, and isolated ExbBD complexes show ion conductivity and cation selectivity in vitro (*Celia et al., 2016*). The ExbBD complex can be thought of as the engine and TonB a drive shaft connecting the engine to the gate in the outer membrane.

Other channel complexes such as MotAB/PomAB (*Kojima, 2015*), TolQR (*Lloubès et al., 2012*; *Godlewska et al., 2009*), and AglRQS (*Agrebi et al., 2015*) exhibit sequence homology to ExbBD and also utilize ion motive force, but have different physiological functions. MotAB and PomAB generate torque for flagellar rotation, TolQR maintains outer membrane integrity, and AglRQS energizes surface gliding and sporulation in slime bacteria. Although structural information on this family is limited, recently crystal structures of the ExbB pentamer with and without a short segment of ExbD including the TM region have been solved, giving valuable insights into the architecture of the ExbB complex. However, a pentameric functional unit seems not to fit with previous studies (*Higgs et al., 2002*; *Jana et al., 2011*; *Pramanik et al., 2011*; *Sverzhinsky et al., 2014*; *Sverzhinsky et al., 2015*).

Here, we report the structures of complexes of hexameric ExbB and ExbD revealed by X-ray crystallography and single particle cryo-EM. Image analyses show hexameric and pentameric complexes coexisting in detergent micelles and within lipid bilayers. Formation of the hexamer is promoted at

higher pH, in line with an increased macroscopic channel conductance. The architecture of the ExbB hexamer and ExbD trimer complex suggests a model for utilization of PMF.

## Results

### Crystal structure determination

We expressed and purified *Escherichia coli* full-length ExbB-ExbD complexes. The structure of crystals grown at pH 9.0 was solved to 2.8 Å by molecular replacement. A starting hexamer model was constructed by docking the coordinates of the ExbB monomer cut out from the pentamer (*Celia et al., 2016*) into a low-resolution cryo-EM map of the hexameric complex (See Materials and methods). The asymmetric unit in the lattice with space group $P2_1$ (*Supplementary file 1*-Table S1) contains two ExbB hexamers, one disposed upside down with respect to the other, and interacting laterally with each other (*Figure 1—figure supplement 2*). The overall hexamer structure resembles a bell with a vertical dimension of ~110 Å and a maximal horizontal dimension on the cytoplasmic side of ~85 Å (*Figure 1*). Throughout this paper 'top' and 'bottom' correspond to the periplasmic and cytoplasmic ends, respectively. We also define the periplasmic side 'up' and the cytoplasmic side 'down' and describe the structure accordingly.

The proportion of ExbD tended to decrease upon crystallization, as deduced from SDS-PAGE patterns of crystal samples (*Figure 1—figure supplement 3*). Yet, peptide mass finger printing analysis showed that some ExbD is retained in the crystals. A Fourier difference map revealed untraceable masses, probably parts of ExbD TM helices, inside the channel of the hexamer. Low occupancy and possibly flexibility may account for the lack of clarity.

### Comparison with the pentamer structure

The ExbB monomers in the hexamer show similar structures to those in the pentamer (the average RMSD of $C\alpha$ atoms between two monomers is 1.4 Å), despite the different subunit number (*Figure 2A*). The structure has been described previously (*Celia et al., 2016*), and, briefly, consists of three long $\alpha$-helices (80 ~ 100 Å), one of which is broken at its middle in the cytoplasm, extending across the cytoplasmic and periplasmic spaces, and connected with short $\alpha$-helices and loops; in total 7 $\alpha$-helices ($\alpha 1$ - $\alpha 7$ from the N-terminus) are in each monomer. The TM region is well conserved over species (*Figure 2A*).

The hexamer and pentamer form similar funnel structures with large cytoplasmic and smaller periplasmic domains connected through the TM region (*Figures 1* and *Figure 2—figure supplement 1*). The overall volume is ~60,400 Å$^3$ in the hexamer with an outermost diameter of ~85 Å on the cytoplasmic side and ~43,900 Å$^3$ and ~75 Å respectively in the pentamer. The channel at center is surrounded by TM helices $\alpha 6$ and $\alpha 7$ in the membrane region and $\alpha 5$ and $\alpha 7$ further down (*Figures 1*, *2A* and *Figure 2—figure supplement 1*). The channel in the hexamer is significantly larger than that in the pentamer (*Figure 1D*). The diameter in the hexamer is ~17 Å and that in the pentamer ~10 Å through the TM region. The pentamer structure of the crystal grown at pH 4.5 resolved one rod-like density located in the central channel towards the periplasm, and a part of the rod was assigned to a TM segment of ExbD (*Celia et al., 2016*). The channel of the pentamer can accommodate only one helix, whereas that of the hexamer has multiple helices. This is confirmed in cryo-EM maps seen below. The larger central channel in the hexamer would fit better with a multimerizing nature of ExbD (*Higgs et al., 2002*; *Ollis et al., 2009*; *Gresock et al., 2015*).

In both complexes, the large cytoplasmic domain has a cluster of positively charged residues surrounding the channel close to the membrane surface and a group of negatively-charged residues at the end on the cytoplasmic side (*Figure 2—figure supplement 1C*). Neighboring monomers of ExbB in the hexamer and in the pentamer bury approximately the same surface areas (3000 Å$^2$) and the residues contributing to interactions are also similar. It is not obvious why hexamers crystallized at pH 9.0, but the negatively charged amino acids in the acidic cluster at the cytoplasmic end are further apart in the hexamer, and the channel radius is particularly large here (*Figures 1D*, *2B*, *Figure 2—figure supplement 1A and B*). Lower pH increases chances for hydration of ionised acidic residues, which may allow the denser packing here of the pentamer. This region is important for function, as deletion of residues 100–109, which include four acidic residues (Asp 102, Asp 103, Glu 105, and Glu 109) decreases iron transport activity (*Bulathsinghala et al., 2013*). It should be

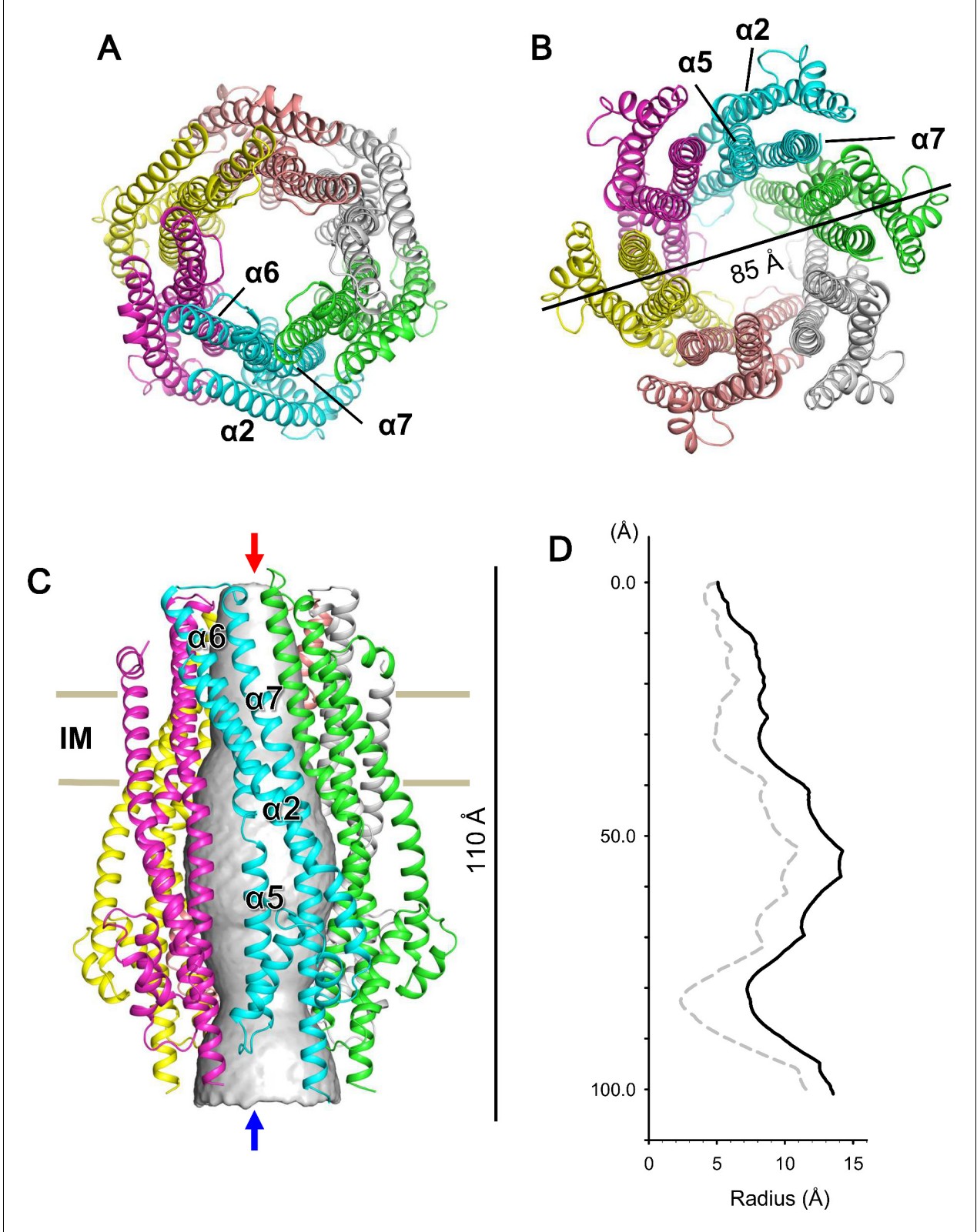

**Figure 1.** Crystal structure of the ExbB hexamer. (**A**) Viewed from the periplasmic side indicated by a red arrow in (**C**). (**B**) From the cytoplasmic side indicated by a blue arrow in (**C**). (**C**) Viewed parallel to the membrane. Overlaid with volume rendering (*Smart et al., 1996*) inside the central channel. Long α-helices are labelled α2, α5, α6 and α7 for one subunit in cyan. See also *Figure 2A*. (**D**) Plots of the channel radii (*Smart et al., 1996*) of the hexamer (solid line) and pentamer (dashed line; *Celia et al., 2016*).

*Figure 1 continued on next page*

*Figure 1 continued*

DOI: https://doi.org/10.7554/eLife.35419.003

The following figure supplements are available for figure 1:

**Figure supplement 1.** Schematic diagram of TM transport mediated by the Ton system.

DOI: https://doi.org/10.7554/eLife.35419.004

**Figure supplement 2.** Crystal packing and electron density maps of the ExbB hexamer.

DOI: https://doi.org/10.7554/eLife.35419.005

**Figure supplement 3.** Characterization of purified ExbB and ExbD and crystals grown at pH 9.0.

DOI: https://doi.org/10.7554/eLife.35419.006

noted that in the previous structure study, well-diffracting crystals of pentamers at pH 7.0 or lower were obtained following treatment of ExbB with a methyl-reducing reagent to modify the amide in lysine side chains (*Celia et al., 2016*). Indeed, the modified lysine near Glu 109 and a calcium ion bound to all five Glu 105 side chains appeared to be needed for pentamer formation in the crystals (*Figure 2d* of *Celia et al., 2016*). Our crystals do not need such treatment.

## Coexistence of the hexameric and pentameric complexes

We measured the channel conductance of ExbBD complexes reconstituted into planar lipid bilayers. The behaviour of the channel is similar to that reported in *Celia et al., 2016*: The conductance shows high pH dependence, poor at low pH and greater at neutral pH or above (*Figure 3*; *Celia et al., 2016*).

We then examined frozen-hydrated specimens of ExbBD complexes (*Figure 4—figure supplement 1A and B*) purified at various pHs with cryo-EM. Reference-free 2D class average of molecular images produced characteristic projections (*Figures 4A, B*, *Figure 4—figure supplement 1B and E*). They include hexameric and pentameric structures corresponding to projections normal to the membrane plane. The hexameric and pentameric images resolve axial projections of the α-helices and well resemble the crystal structures viewed from the same direction (*Figure 4A and B*). Complexes of ExbB pentamer and hexamer coexist in the sample solution and the proportion of each depends on pH, being 3:1 at pH 5.4, 1:3 at pH 8.0, and almost entirely hexamers at pH 9.0 (*Figure 4C* and *Supplementary file 1*-Table S2).

The number of ExbB and ExbD subunits in the functional unit has been argued for years (*Higgs et al., 2002*; *Jana et al., 2011*; *Pramanik et al., 2011*; *Sverzhinsky et al., 2014*; *Sverzhinsky et al., 2015*). Estimation of the subunit number has been elusive likely due to this pH dependent pentamer/hexamer equilibrium. A previous MS analysis showed that the main fraction of the sample in the weak basic condition used corresponded to ExbB hexamers and small amounts of the pentamer were also found (*Pramanik et al., 2011*). This proportion is consistent with our results (*Figure 4*).

Hexamers and pentamers also coexist within lipid bilayers. We prepared 2D crystals at pH 5.4 by reconstituting purified ExbBD complexes into liposomes. Fourier transform of the crystals showed pseudo-hexagonal lattice patterns as in 3D crystals in *P*1 lattices of another crystal form found by X-ray diffraction experiments (See Materials and methods). Fourier filtering of different patches from negatively-stained 2D crystals revealed pentagonal and hexagonal images (*Figure 5*). The coexistence of pentamers and hexamers probably prevents growth of well-diffracting crystals at pH 5.4 (see Materials and methods).

The 2D crystals contained both full-length ExbB and ExbD (*Figure 5E*). In contrast, loss of ExbD occurred in 3D crystals (*Figure 1—figure supplement 3C*). The 3D crystals appeared to be built by stacking of 2D crystal layers (*Figure 1—figure supplement 2A and B*). The periplasmic part of ExbD, which includes a long loop and a compact peptidoglycan-binding motif, would be difficult to fit between the 2D crystal layers in both *P*2$_1$ and *P*1 3D crystal lattices without deforming these domains (*Figure 1—figure supplement 2A and B*). Thus, stacking of 2D crystal layers might exclude ExbD because of these steric effects.

The measurement of the single channel conductance showed that there are two major conductance states at both pH 7.5 and at pH 4.5, exhibiting conductances of 140–150 pS and ~60 pS (*Figure 3A and B*). The frequency of the high conductance state was about 2.6 times that of the low

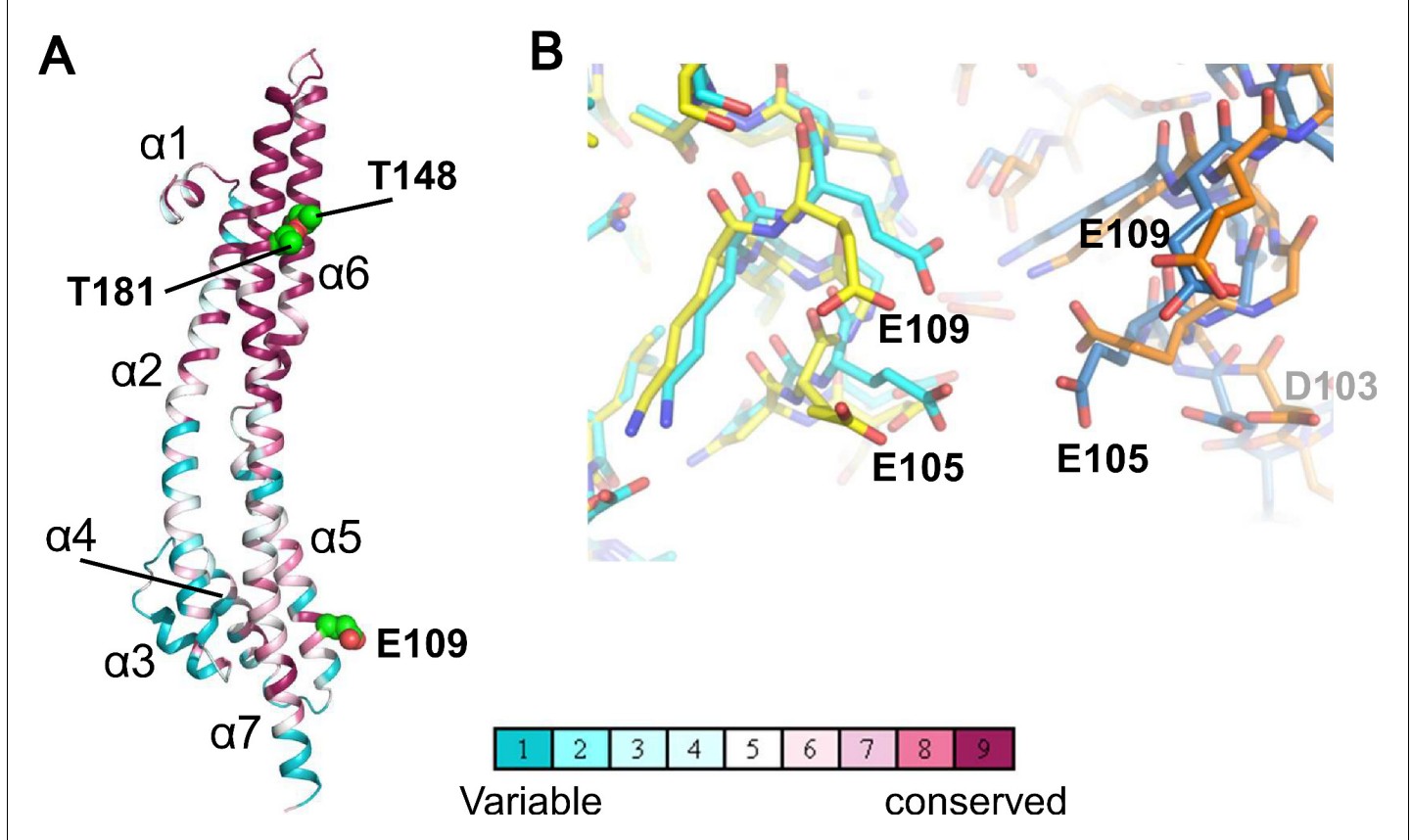

**Figure 2.** Structure of the ExbB monomer and interface between two subunits at the cytoplasmic end in the crystal. (A) ExbB monomer in gradient colour representation for conserved and variable amino acids (*Landau et al., 2005*). Sequences were compared over 150 homologous proteins from the Uniprot database and aligned with BLAST. Thr 148 and Thr 181 face inside the channel. These threonines and Glu 109 in the acidic cluster (B) are displayed in CPK representation. (B) The acidic cluster between two neighbouring subunits in the cytoplasmic end. Viewed from inside the channel. See also *Figure 2—figure supplement 1C*. Stick representations for subunits of the hexamer in yellow and orange, and those of the pentamer in cyan and sky blue.

DOI: https://doi.org/10.7554/eLife.35419.007

The following figure supplement is available for figure 2:

**Figure supplement 1.** Side by side comparison of the crystal structure of the ExbB hexmer and pentamer, and electrostatic surface of the hexamer.
DOI: https://doi.org/10.7554/eLife.35419.008

at pH 7.5 (47 events versus 18), and was about half that of the low at pH 4.5 (19 events versus 37). Thus two distinct states co-exist and frequency distributions depend on pH, tying in with the EM data if the hexamer and pentamer represent channels with high and low conductance, respectively. The two states/channels appear interchangeable in the lipid bilayer as macroscopic currents are reversible on adjusting pH (*Figure 3C and D*).

## Cryo-EM structures

As shown, the electron density map of the crystal of the ExbB hexamer and ExbD complex does not allow modelling of ExbD (*Figure 1—figure supplement 2C and D*), even though the crystal retains a small amount of ExbD (*Figure 1—figure supplement 3C*). Purified samples actually contain a significant amount of ExbD (*Figure 1—figure supplement 3B*). To identify ExbD in the complex, we carried out 3D reconstruction from the cryo-EM images of the samples at pH 8.0 by single particle analysis (*Supplementary file 1*-Table S2). Starting from a reference map filtered to 30 Å resolution from the atomic model of the ExbB hexamer, a 3D structure was determined at 6.7 Å resolution (*Figure 4—figure supplement 1C*) with Bayesian-based algorisms implemented in RELION (*Scheres, 2012*). The heterogeneity (*Figure 4B*) as well as the relatively low molecular weight and

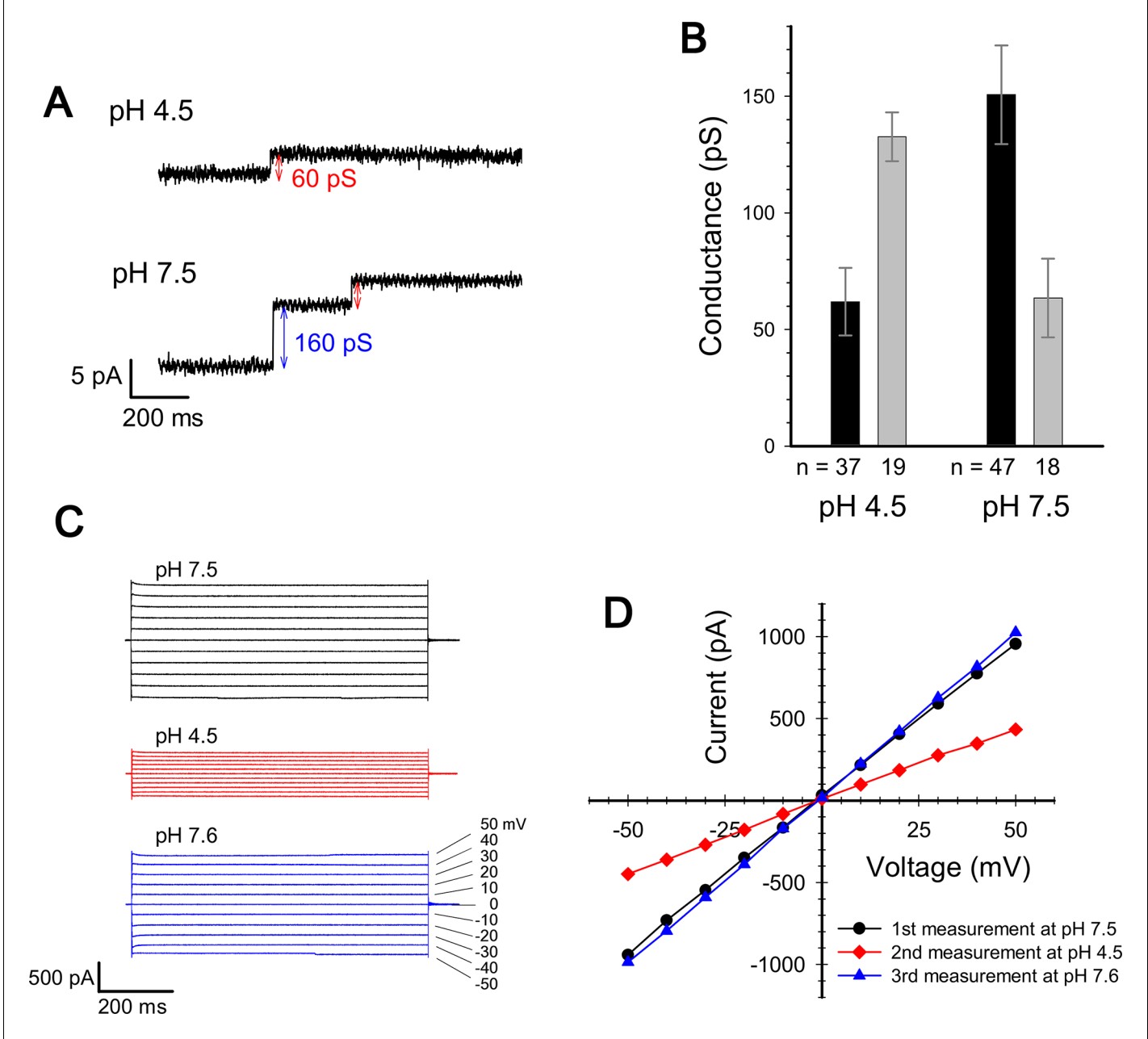

**Figure 3.** Current measurement for ExbBD complexes in planar lipid bilayers. (A) Typical charts of channel current at pH 7.5 and pH 4.5 with a voltage of ± 50 mV. (B) The single channel conductance at pH 4.5 and 7.5. Two major modes are shown with number of observations (n). Incremental points as in (A) were counted as the event of the single channel, as it is unlikely that two events occur simultaneously. (C) Macroscopic current traces upon voltage steps for many ExbBD complexes in the planar bilayers. The current was first measured at pH 7.5 (black) and then the solutions in the cis and trans chambers were changed to pH 4.5 (red) for another measurement by adding succinic acid. After that, it was returned to pH 7.6 (blue) with Tris buffer. (D) Current-voltage diagrams of the datasets in (C).

DOI: https://doi.org/10.7554/eLife.35419.009

low-symmetry did not allow reconstruction at higher resolutions and densities corresponding to side chains were not visible.

Nonetheless, the map clearly resolves all α-helices and loops in the ExbB hexamer except for small parts of the loops at the periplasmic and at the cytoplasmic ends (***Figures 6*** and ***Figure 6—figure supplement 1A***). Densities corresponding to bound detergent around the TM region are visible at a lower contour level (***Figure 6—figure supplement 1A***). The map also reveals three rod-like

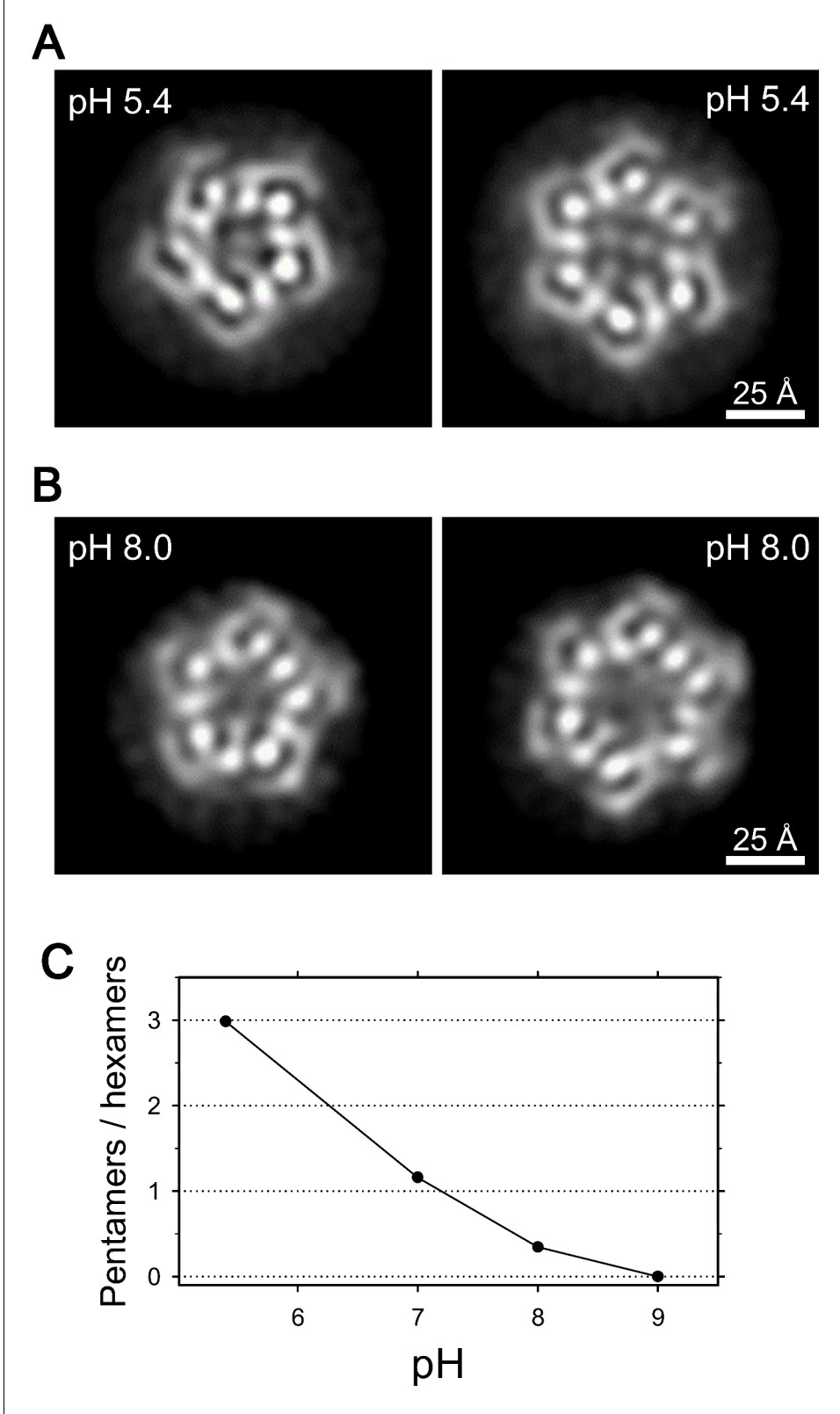

**Figure 4.** Coexistence of hexameric and pentameric ExbBD complexes. (**A**) 2D class average at pH 5.4. (**B**) 2D class average at pH 8.0. (**C**) Plot showing the ratio of particle numbers contributed to pentameric averages against those to the hexameric averages. See also *Supplementary file 1*-Table S2 for the particle numbers.

DOI: https://doi.org/10.7554/eLife.35419.010

*Figure 4 continued*

The following figure supplement is available for figure 4:

**Figure supplement 1.** Cryo-EM images of the ExbBD complex, and resolution estimates.
DOI: https://doi.org/10.7554/eLife.35419.011

densities inside the channel, which likely correspond to the TM helices of three ExbD molecules (*Figure 6*). The helices fit snugly into the hexamer channel, with the arrangement off-centre and asymmetrical. Indeed, the rod-like densities disappear if 3-fold symmetry is enforced. The local resolutions were estimated to be 4 ~ 5.5 Å for most of the α-helices both in the ExbB hexamer and the central channel (*Figure 4—figure supplement 1D*). Several lines of evidence show that ExbD has a propensity for dimerization (*Celia et al., 2016*; *Higgs et al., 2002*; *Ollis et al., 2009*), but can be monomeric as in an NMR experiment at low pH (*Garcia-Herrero et al., 2007*). Also, ExbD forms a heterodimer with TonB (*Ollis et al., 2009*; *Gresock et al., 2015*), further suggesting more than one oligomerization possibility for ExbD.

Another interesting feature in the cryo-EM map is that a single mass in the periplasm becomes visible at ~ a half-contour level (*Figure 6—figure supplement 1B*) of that in *Figure 6—figure supplement 1A*. The mass is ~ 50 Å above the membrane surface. ExbD is predicted to consist of a compact periplasmic domain, one TM helix, and a loop region connecting these two. The compact periplasmic domain in the NMR structure of ExbD consists of several short β-strands and α-helices facing each other, and this arrangement is seen in other peptidoglycan binding proteins (*Garcia-Herrero et al., 2007*; *Wojdyla et al., 2015*). We docked this domain of ExbD in the cryo-EM map, and the dimension of the mass in the periplasm is such that it accommodates three domain structures well (*Figure 6—figure supplement 1B*).

We also reconstructed a 3D structure of the pentameric complex at pH 5.4 in the same way. The 3D map resolved a pentameric structure with clear three long α-helices per subunit in the cytoplasmic domain but poorer masses in the TM region except for a rod-like density inside the channel. This is probably due to the lower molecular weight of the complex compared with the hexamer and more preferential orientations for top views in cryo-EM images of samples at pH 5.4. Five-fold symmetrization, however, gave a better map. Although the quality of this pentamer map, particularly the continuity of α-helices at the periplasm, is still not quite as good as those of the hexameric complex, a single rod-like density appears again inside the channel in the TM region (*Figure 6—figure supplement 1C*). The rod probably corresponds to the TM helix of ExbD. There is insufficient space to accommodate more helices in the channel. The findings are consistent with the crystal structure of the pentamer at pH 4.5 (*Celia et al., 2016*).

## Structural changes in the single particle reconstructions

The crystal structure of the ExbB hexamer needed slight modifications to be fitted into the cryo-EM map of the hexameric complex (*Figure 6*). This was done by rigid-body refinement of the atomic model of the ExbB hexamer and three ExbD TM helices (ExbD$_{TM}$) against the cryo-EM map in real space (*Supplementary file 1*-Table S3). In the crystal, each ExbB subunit is settled on a flat plane parallel to the membrane and well follows the six-fold symmetry normal to this plane. In contrast, each subunit in the fitted model (ExbB$_6$ExbD$_{3TM}$) is gradually dislocated by 1 ~ 3 Å from the membrane plane downwards towards the cytoplasmic side in the cryo-EM structure (*Figure 6C*).

The arrangement of the ExbB models in the ExbB pentamer with one ExbD$_{TM}$ (ExbB$_5$ExbD$_{1TM}$) shows little change from the crystal structure (*Figure 6—figure supplement 1C*).

## Discussion

### Ion pathway

Our results show that pentameric and hexameric complexes coexist in detergent micelles and within lipid bilayers (*Figures 4* and *Figure 4—figure supplement 1*), and the ratio of the hexamer to the pentamer increases with pH (*Figure 4* and *Supplementary file 1*-Table S2), similar to the pH dependence of the macroscopic channel conductance (*Figure 3*; *Celia et al., 2016*). This suggests that the pentamer is less active and the hexamer more active. The crystal structure at pH 4.5 (*Celia et al.,*

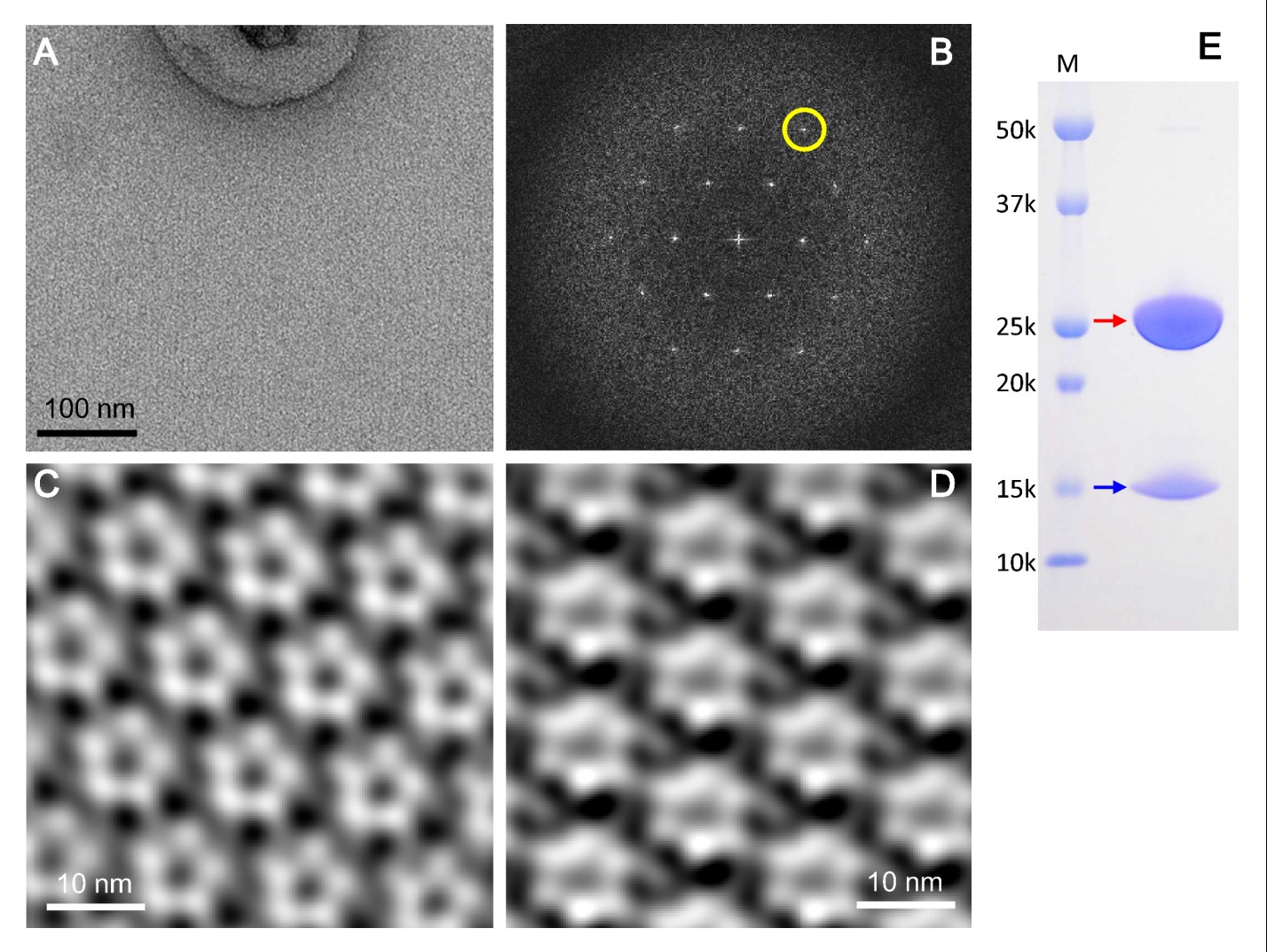

**Figure 5.** 2D crystals of the ExbBD complex in lipid bilayers. (**A**) A representative EM image of a negatively-stained 2D crystal. (**B**) Fourier transform of a crystal patch. A pseudo-hexagonal lattice pattern is visible. A yellow circle refers to a diffraction spot at ~35 Å resolution. (**C**) A typical array of pentagonal molecules obtained by Fourier-filtering of crystal patches. (**D**) A typical array of hexagonal molecules by Fourier-filtering other crystal patches. All the figures in (**A**), (**B**), (**C**) and (**D**) were produced from 2D crystals grown at pH 5.4. Eleven images were analysed. (**E**) A SDS-PAGE pattern of 2D crystals. Red and blue arrows correspond to ExbB and ExbD, respectively. Lane marked 'M' indicates marker.
DOI: https://doi.org/10.7554/eLife.35419.012

*2016*) and our cryo-EM reconstruction of the pentameric complex (*Figure 6—figure supplement 1C*) show a central single TM helix of ExbD, which makes a ~2 Å pore at the shortest diameter (*Figure 7C*). The ExbBD complex has been shown to be a cation-selective channel (*Celia et al., 2016*), but this pore size is too small for permeation of cations unless there are large fluctuations of the central helix. Thus, the structure of this pentameric complex suggests a low efficiency, blocked or semi-blocked, energy transducing state. The ExbB hexamer and ExbD trimer complex has a ~5.5 Å pore (*Figure 7C*), which fits with the reported pore size of cation channels.

The inner surface of the hexameric subunits of ExbB in the TM region is lined with hydrophobic residues and polar residues such as Thr 148 and Thr 181 (*Figures 2A*, *Figure 1—figure supplement 2C and D*). A thorough mutational study has indicated that there are no critical residues in ExbB directly involved in proton transport (*Baker and Postle, 2013*). In the case of ExbD, Asp 25 in the lower part of the TM helix is critical for transport mediated by the Ton system (*Braun et al., 1996*), and conserved over this family, and is likely part of the proton conductance pathway. Significantly,

the cryo-EM structure shows that each of the subunits of the hexamer are slightly shifted downwards with respect to the prior adjacent molecule, resulting in a downward spiral of the hydrophobic ExbB wall that leads to the Asp 25 residues of the trimer (*Figure 7D*). These aspartates may form a cation selective filter (*Celia et al., 2016*). In our model, the aspartates reside at the lower part of the TM region and two of them face the pore, producing an electronegative field in the ion path (*Figure 7B*). The aspartates likely allow only cations to pass, and those cations may be funnelled by the electropositive circle of residues lower down and propelled towards the electronegative toroid at the exit (*Figures 7B* and *Figure 2—figure supplement 1C*).

## A model for utilization of proton motive force

The asymmetric disposition of the three TM helices of ExbD within the channel may be important for utilizing PMF. Many biological molecular machines depend on asymmetry. Vacuolar-type ATPase is an example, where a part of the single c' subunit is located off-centre of the symmetric integral membrane c-ring, resulting in rotation and proton pumping using the energy of ATP hydrolysis (*Mazhab-Jafari et al., 2016*). A similar mechanism may operate in the ExbBD complex, as dynamic motion of TonB, possibly rotation, has been observed by fluorescent optical microscopy (*Jordan et al., 2013*). The proton gradient through the spiral of ExbB subunits could generate a torsional force (*Chang et al., 2001*) to induce rotational movement of the ExbD TM helices relative to the ExbB ring (*Figure 8A*) and step-wise participation of two of the three Asp 25 residues as they become exposed in the channel (*Figures 6C* and *7B and D*). The force could be transduced to TonB – it is known that ExbD interacts with TonB through their periplasmic domains (*Ollis et al., 2009*; *Gresock et al., 2015*). TonB resides around the ExbB complex, as TonB TM residues interact with the outward-facing side of the first TM helix (α2) of ExbB (*Figures 1* and *6C*; *Larsen et al., 1994*; *Larsen et al., 1999*; *Larsen and Postle, 2001*). We have tried to obtain stable complexes of TonB-ExbB-ExbD, TonB-ExbB and TonB-ExbD for structural studies, but without success. The difficulties may reflect the peripheral location of TonB, outside of the hexameric complex, and a tendency to dissociate from it. In this model, TonB would have little effect on the complexation of ExbB and ExbD (*Celia et al., 2016*). TonB may move around the outer rim of the hexamer, and also promote lateral movement of the whole TonB complex in the inner membrane through transient contact of the periplasmic C-terminal domain of TonB with the PG layer as proposed by *Klebba, 2016*. This movement could facilitate linking up with outer membrane receptors (*Klebba, 2016*) to induce a conformational change that releases receptor-bound substrate into the periplasmic space (*Chang et al., 2001*; *Klebba, 2016*).

## Oligomeric variety and transformation

The equilibrium of hexamer and pentamer may depend on the protonation state of acidic residues such as Glu 105, Glu 109, Asp 103, Asp 225 at the cytoplasmic end of the channel (*Figures 2B* and *7D*), in accordance with PMF. If the hydronium concentration rises in this vicinity, protonation of these residues and easing of electrostatic repulsion may allow denser packing among ExbB subunits, which repels one ExbB subunit from the hexamer ring and excludes two ExbD from the central space to form the pentamer ring (*Figure 8B*). Reversely, a hydronium concentration decrease and deprotonation of the acidic residues would shift to looser packing, which recruits two ExbD to the ring and one ExbB to form the hexameric complex. The channel in the hexameric complex accommodates three but not four TM helices. Also, a lower number of TM helices in the hexamer channel, as well as a vacant pentamer channel, produce larger pores and unproductive and high proton influx. Thus, the channel activity may be controlled by joining/ejecting two ExbD molecules and one ExbB through partial complex disruption. The ExbD dimer is stable (*Celia et al., 2016*; *Jana et al., 2011*; *Ollis et al., 2009*) and excluded ExbB molecules may merge into smaller multimers according to cross linking (*Jana et al., 2011*). The low conductance of the pentamer would not only stop import of compounds, but decreased proton influx could also prevent excessive hydronium accumulation within the cytoplasm of the bacterium (*Figure 8B*).

Oligomeric variation is displayed by the mechano-sensitive channel, MscL (*Walton et al., 2015*) and other homologous oligomeric membrane proteins such as $F_0$-ATPases (*Watt et al., 2010*) and light harvesting complexes (*Cogdell et al., 1997*). It is, however, not clear whether these various oligomeric states relate to function or are artefacts due to detergent solubilisation, or deletion

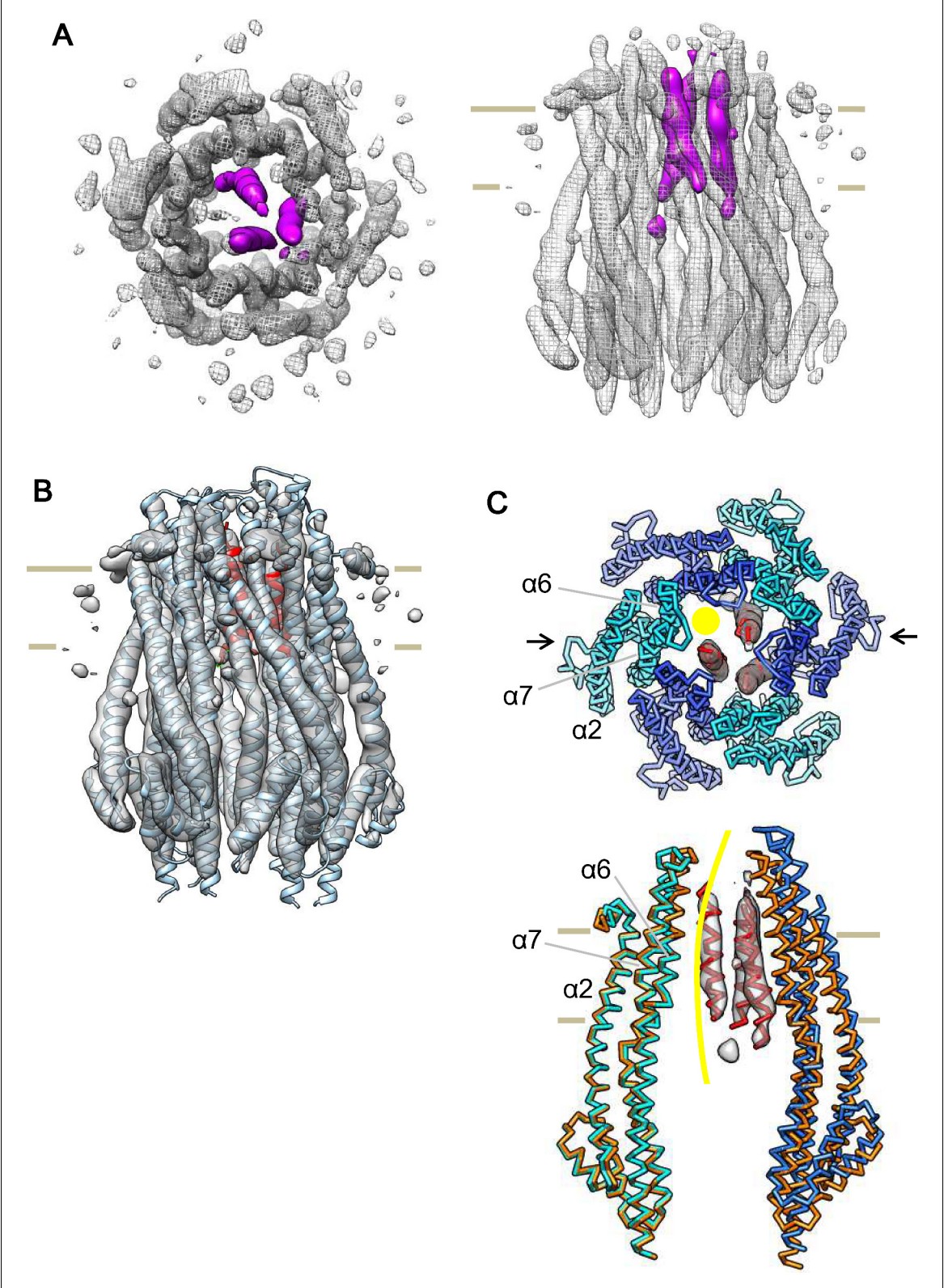

**Figure 6.** Cryo-EM structure of the ExbB hexamer and ExbD trimer complex. (**A**) Top and side views of the hexameric complex. Reconstructed to 6.7 Å resolution from cryo-EM images of samples at pH 8.0. The arrangement of the ExbD helices coloured in magenta is asymmetric inside the channel. (**B**) The same side view as in (**A**). Overlaid with the atomic models of ExbB in cyan and ExbD TM helices (ExbD$_{TM}$) in red. The model of ExbB$_6$D$_{3TM}$, ExbB subunits in the hexamer crystal and ExbD$_{TM}$ were fitted into the cryo-EM reconstruction by rigid-body refinement in real space (**Adams et al., 2010**).
*Figure 6 continued on next page*

*Figure 6 continued*

(C) Structure changes in the ExbB$_6$D$_{3TM}$ model. The cryo-EM map corresponding to ExbD$_{TM}$ is displayed. Side view only shows two ExbB subunits in cyan and sky blue (indicated with arrows in the top view in **C**) overlaid with the same ones in the crystal (orange). The subunit on the left is superimposed on that of the crystal and that on the right shows the dislocation from the position of the crystal subunit needed to fit the cryo-EM map. The viewing direction of the side view in (**C**) is rotated anticlockwise by ~40° around the channel axis from those in (**A**) and (**B**). A yellow circle in the top view and curve in the side view indicates the putative ion pathway.

DOI: https://doi.org/10.7554/eLife.35419.013

The following figure supplement is available for figure 6:

**Figure supplement 1.** Cryo-EM structures of the hexameric and pentameric ExbBD complexes.

DOI: https://doi.org/10.7554/eLife.35419.014

mutation (*Walton et al., 2015*). Unlike other examples, we observe formation of hexamer and pentamer complexes in detergent micelles and within lipid bilayers, and record two distinct ion conductivities in a pH dependent ratio in planar lipid membranes for the same ExbBD sample. These characteristics could be relevant to the biological nature of the energizer in vivo. Oligomeric transformation may be facilitated by a highly fluid lipid membrane. Indeed, high fluidity allows diffusion of MotB molecules within the cell membrane (*Leake et al., 2006*) and expedites large movements of TM helices in Ca$^{2+}$-ATPase during the reaction cycle (*Norimatsu et al., 2017*). The observations reported here may provide a new vision of dynamic membrane biology and spur further studies to explore the processes involved.

## Materials and methods

### Cloning, co-expression and purification of the ExbBD complex

The *E. coli exbB* and *exbD* genes were subcloned into vector pET-21b (Novagen). A hexahistidine tag was added at the C-terminus of ExbB, but not on ExbD. The plasmid was transformed in *E. coli* BL21 (DE3) Gold cells (Novagen), and the cells were grown at 37°C in LB medium. To induce ExbB and ExbD, IPTG was added to a final concentration of 0.5 mM when the culture reached an optical density at 600 nm (OD$_{600nm}$) of ~1.0. After additional culture for 3 hr at 30°C, the cells were harvested by centrifugation at 4000 g for 30 min at 4°C. The pellet was resuspended in 50 mM Tris-HCl, pH 8.0, 0.3 M NaCl, 10% (v/v) glycerol, 10 mM imidazole, protease inhibitor cocktail cOmplete (Roche), DNase I (Roche), 5 mM MgCl$_2$, and 0.2 mg/ml lysozyme and disrupted with ~7 passages at ~10,000 psi through a high pressure homogenizer EmulsiFlex C-5 (AVESTIN). Cell debris was removed by centrifugation at 10,000 g for 30 min at 4°C, and the membrane fraction was collected by ultracentrifugation at 125,000 g for 90 min at 4°C.

The ExbBD complex was purified by immobilized metal affinity (IMAC) and size-exclusion chromatography (SEC). The membrane fraction was first solubilized with 1–2% (w/v) n-dodecyl-β-D-maltoside (DDM; Dojindo) in 50 mM Tris-HCl, pH 8.0, 0.3 M NaCl, 10 mM imidazole, 10% glycerol, and cOmplete. The insoluble membrane fraction was removed by centrifugation at 125,000 g for 60 min at 4°C. The supernatant was then loaded onto a column filled with a metal affinity resin COSMOGEL His-Accept (Nacalai tesque) at 4°C. The column was washed with the solubilization buffer except for containing 0.018% (w/v; × 2 critical micelle concentration; cmc) DDM, and developed with a stepwise gradient of imidazole concentration. Fractions eluting at ~0.3 M imidazole were collected and concentrated using a spin concentrator with a molecular weight cut-off of 100,000. The concentrate was then applied to a SEC column Superdex 200 10/300 GL (GE Healthcare), and run in 10 mM glycine, pH 9.0, 0.2 M NaCl, 10% glycerol, 1 mM TCEP, 0.5% (v/v; × 2 cmc) tetraethylene glycol monooctyl ether (C$_8$E$_4$; Anatrace) or 0.07% (v/v; × 2 cmc) pentaethylene glycol monodecyl ether (C$_{10}$E$_5$; Anatrace). The peak was collected and the purity was checked by SDS-PAGE. The protein concentration was measured by the BCA assay (Pierce).

We also prepared a construct of the ExbBD complex with a TEV protease recognition site (Glu-Asn-Leu-Tyr-Phe-Gln-Gly-Ser) inserted between the C-terminus of ExbB and a hexahistidine tag. The protein complex was induced and purified in the same way except for cleavage of the histidine tag by 3 hr incubation with TEV protease at r.t. Then, the sample, passing through a desalting column to

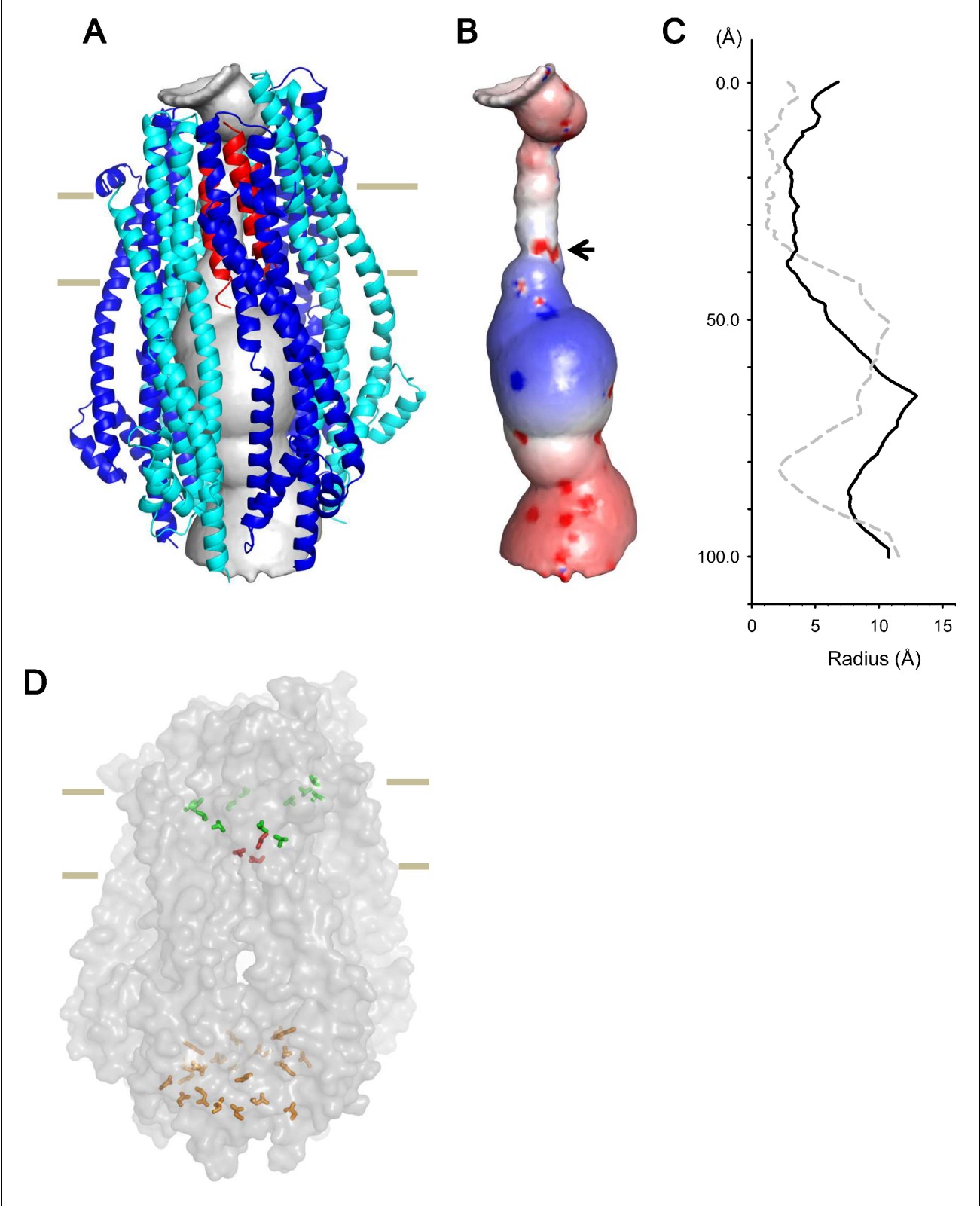

**Figure 7.** Model of the hexameric ExbBD complex. (A) A structure model of the ExbB hexamer and the TM helices of ExbD. Overlaid with volume rendering (**Smart et al., 1996**) inside the channel. (B) Electrostatic surface of the channel is displayed with electropositivity in blue and electronegativity in red. An arrow indicates electronegative surface of an ion-selective filter formed by Asp 25. (C) Plots of the channel radii of the model in (A) (solid line) and the pentamer with a ExbD helix in the crystal (dashed line; **Celia et al., 2016**). The central helix in the pentamer crystal is shifted to the periplasmic

*Figure 7 continued on next page*

*Figure 7 continued*

side (upper; *Celia et al., 2016*). (D) A side view of ExbB$_6$ExbD$_{3TM}$ in surface rendering. The side chains of Thr 148, Thr 181 (green) in ExbB and Asp 25 (red) in ExbD are displayed as markers. The side chains of Asp 103, Glu 105, Glu 109, and Asp 225 are also displayed at the cytoplasmic end of the channel in orange. As the ExbB subunits are gradually shifted downwards towards the cytoplasm, residues surrounding the channel such as Thr 148 and 181 follow a spiral downward path.

DOI: https://doi.org/10.7554/eLife.35419.015

remove imidazole, was mixed with the metal affinity resin and stirred for 1 hr at 4°C. The unbound fraction was run on SEC with the buffer.

## Crystallization and protein identification

Samples were concentrated to ~10 mg/ml and subjected to extensive screening over sparse matrix conditions with a Mosquito crystallization robot (TTP Labtech). Crystals were grown by hanging-drop vapor diffusion at 20°C in a mother liquor containing 0.1 M glycine, pH 9.0, 0.15 M CaCl$_2$, ~40% PEG 350 MME and 0.05–0.2 M L-arginine. Plate-like crystals of approximately 100 μm × 100 μm × 10 μm grew over 1–2 months. Hexagonal crystals grew in mother liquors at pH 5.4.

To identify the crystal content, crystals were collected with a nylon loop and washed with the crystallization buffer. The crystals were dissolved in SDS sample buffer, heated at ~95°C for 5 min, and then run on a 15% homogeneous polyacrylamide gel. Bands cut out from the gel stained with Coomassie brilliant blue were digested with trypsin. The digestion mixtures were subjected to MALDI-LIFT TOF/TOF MS on an Ultraflextreme mass spectrometer (Bruker Daltonics) and/or LC MS/MS on a Q Exactive (Thermo Fisher Scientific) followed by peptide mass finger printing analysis. The obtained MS spectra were used to search the SwissProt database using the Mascot program (*Perkins et al., 1999*). ExbB and ExbD were identified with remarkably high identity scores. No other candidates were flagged in the database.

## Data collection and structure determination by X-ray crystallography

Crystals were flash-frozen in liquid nitrogen and diffraction data were collected on the synchrotron radiation beam at BL26, BL32XU and BL41XU of SPring-8 at a wavelength of 1 Å. The datasets from crystals grown at pH ~9 were processed with XDS (*Kabsch, 2010*). The crystals belonged to space group *P*1 or *P*2$_1$. Crystals grown at pH 5.4 yielded anisotropic patterns with high diffuse scattering backgrounds, where diffraction spots were limited to 7–8 Å resolution from their hexagonal crystal plane. The datasets from these crystals could not be processed.

To phase the diffraction data, we used a cryo-EM map and the atomic coordinates of the ExbB pentamer (PDB accession code: 5SV0; *Celia et al., 2016*). The atomic model of the ExbB monomer cut-out from the pentamer was fitted into a low-resolution cryo-EM map of the hexamer (see below) using UCSF Chimera (*Pettersen et al., 2004*). Starting from the constructed hexamer model, molecular replacement was carried out using PHASER (*McCoy et al., 2007*). The solutions were unique and of high scores, achieved for the diffraction data from both crystals in *P*1 and in *P*2$_1$ lattices (*Supplementary file 1*-Table S1). The asymmetric unit contains two ExbB hexamers disposed upside down with respect to each other. The two hexamers interacted mostly through the cytoplasmic end in the *P*1 lattices along the c-axis, but laterally through long α-helices over the TM and cytoplasmic domains in the *P*2$_1$ lattice (*Figure 1—figure supplement 2A and B*). The latter probably made the crystal packing tight. Since crystals in the *P*1 lattices showed incomplete data statistics (*Supplementary file 1*-Table S1) and poorer isomorphism, structure refinement was carried out only against the data from the crystals in the *P*2$_1$ lattice. The structure of the ExbB hexamer was refined using BUSTER (*Bricogne et al., 2016*), and manually adjusted with COOT (*Emsley et al., 2010*). The electron density map resolved amino acids 10–234 (subunit A), 20–234 (subunit B), 10–232 (subunit C), 19–233 (subunit D), 11–232 (subunit E) and 19–234 residues (subunit F) for one hexamer, and 10–234 (subunit G), 19–234 (subunit H), 10–232 (subunit I), 20–233 (subunit J), 11–234 (subunit K) and 20–234 residues (subunit L) for the other. In σ$_A$-weighted 2 |F$_{obs}$| - |F$_{calc}$| and |F$_{obs}$| - |F$_{calc}$| maps, significant densities appeared in the central channel, but assignment of any model for ExbD was not possible. Data collection and refinement statistics are shown in *Supplementary file 1*-Table S1. The

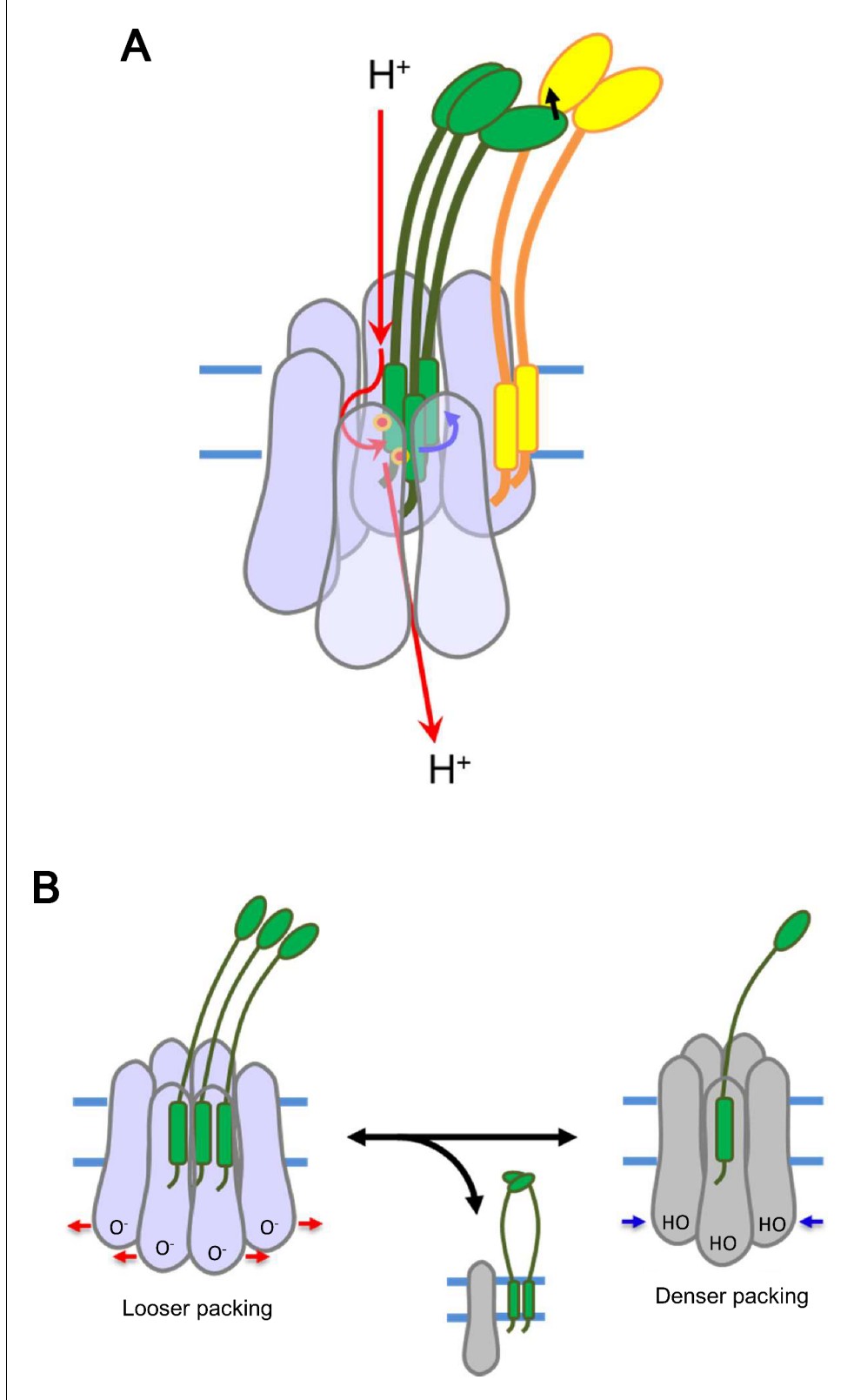

**Figure 8.** Models for energy conversion of the ExbBD complex and activation/inactivation. (**A**) Energy transduction mechanism. Protons coming from the periplasmic funnel could spiral down around the ExbB wall as indicated red arrows. The spiral flux could generate a torsional force (a blue arrow; *Chang et al., 2001*) to induce rotational movement of the ExbD TM helices relative to the ExbB ring. ExbD interacts with TonB through their periplasmic domains (*Ollis et al., 2009*; *Gresock et al., 2015*) and the force could be transduced to TonB (a black arrow), which resides around the *Figure 8 continued on next page*

*Figure 8 continued*

ExbB complex. TonB may move around the outer rim of the hexamer, facilitating linking up with outer membrane receptors (*Klebba, 2016*). The interaction of the TonB C-terminal domain with the periplasmic domain of the receptor induces conformational changes to release a substrate bound to the receptor into the periplasmic space (*Chang et al., 2001*; *Klebba, 2016*). (**B**) Transition mechanism between the hexameric and pentameric ExbBD complexes. If the hydronium concentration rises in the cytoplasmic funnel including many acidic residues, protonation (HO in the right figure) of these residues and easing the electrostatic repulsion may allow denser packing among ExbB subunits, which repels one ExbB subunit from the hexamer ring and excludes two ExbD from the central space to form the pentamer ring. The ExbD dimer is a stable complex and excluded ExbB molecules may form multimers. Thus, the pentameric complex prevents excessive proton influx into the cell. A hydronium concentration decrease and deprotonation (O⁻ in the left figure) of the acidic residues would shift to looser packing, which recruits two ExbD to the ring and one ExbB to form the hexameric complex. ExbB, ExbD and TonB are displayed in violet or grey, green and yellow, respectively. Small red circles in (**A**) indicate Asp 25 in the ExbD TM helix.

DOI: https://doi.org/10.7554/eLife.35419.016

channels were analysed by HOLE (*Smart et al., 1996*) and buried surface areas and volumes were calculated by PDBePISA (*Krissinel and Henrick, 2007*).

## Sample preparation for cryo-EM SPA

To achieve the optimal contrast of molecular images, we used lauryl maltose neopentyl glycol (LMNG; Anatrace), which is a detergent with a low cmc, and glycerol was not included in the buffer for all experiments. Samples were solubilized with 1% (w/v) LMNG and purified in 0.002% (v/v; $\times$ 2 cmc) LMNG by IMAC and SEC as described. To examine samples at different pH values, SEC was carried out in buffers containing either 10 mM Na-citrate, pH 5.4, 10 mM HEPES, pH 7.0, 10 mM Tris-HCl, pH 8.0, or 10 mM glycine, pH 9.0. The SEC patterns showed a small peak consisting of ExbB and ExbD at a lower molecular weight than those in the main fraction. The corresponding peak became smaller when run at pH 8.0 and disappeared at pH 9.0, which is consistent with an equilibrium of pentamer and hexamer (*Figure 4C* and *Supplementary file 1*-Table S2).

For frozen-hydrated sample preparation, gold was manually sputtered on holey carbon film-coated copper grids (Quantifoil R1.2/1.3, Quantifoil Micro Tools GmbH). This reduced charging of samples in ice and thereby minimized beam-induced movement (*Russo and Passmore, 2014*), when the samples were exposed to the electron beam. Three µl of ExbBD samples at a protein concentration of 0.25 - 0.5 mg/ml were applied onto the grid, deposited manually with filter paper from the reverse side of the grid, and rapidly frozen in liquid ethane with a home-made plunger in a cold chamber with a humidifier. The sample grids were screened with a JEOL-2100 electron microscope (JEOL) equipped with a LaB₆ gun operated at an accelerating voltage of 200 kV.

## Cryo-EM image acquisition

The grids frozen in seemingly good conditions were first examined on Falcon II detectors (FEI) in Titan Krios electron microscopes (FEI) at the Research Center for UHVEM, Osaka University and NeCEN, Leiden University. We found that the modulation transfer function (MTF) of the Falcon II (FEI) was not sufficiently high enough to obtain high-resolution information from molecular images in this sample size. Hence, all the data used for image analysis were collected on the K2 Summit direct electron detector (GATAN) as below.

Images of samples at pH 8.0 and pH 5.4 were collected with a Tecnai Polara scope (FEI) operated at an accelerating voltage of 300 kV at UCSF and a Titan Krios operated at 300 kV in eBIC, Diamond Light Source, respectively. Dose-fractionated images were recorded on a K2 summit in super-resolution counting mode. At eBIC, inelastically scattered electrons were removed through a GATAN Quantum energy filter with an energy slit width of 20 eV. The dose rate was six electrons per pixel per s, the total exposure time was 20 s, and one physical pixel corresponded to 1.22 Å at UCSF. The dose rate was 7.2 electrons per pixel per s, the total exposure time was 6 s, and one physical pixel corresponded to 1.06 Å at eBIC. Each frame accumulated electrons for 0.2 s. Defocus values ranged from 0.6 to 3.1 µm both at UCSF and at eBIC. Collected images were used for 3D reconstruction (*Supplementary file 1*-Table S2).

For only 2D classification analysis in this study, images of samples at pH 7.0 and 9.0 were collected on a K2 summit with a Tecnai Arctica (FEI) operated at 200 kV. The dose rate was 7.5 electrons per pixel per s, the total exposure time was 6.4 s with an accumulation time of 0.2 s for each

frame, and one physical pixel corresponded to 0.99 Å. Defocus values ranged from 0.5 to 4.9 μm (*Supplementary file 1*-Table S2).

We also collected images of the complex at pH 8.0 with the histidine tag cleaved off. Images were collected with a Titan Krios operated at 300 kV in eBIC. The dose rate was ~ 5 electrons per pixel per s, the total exposure time was 10 s with an accumulation time of 0.2 s for each frame, and one physical pixel corresponded to 1.06 Å. Defocus values ranged from 0.6 to 2.9 μm (*Supplementary file 1*-Table S2).

## Structure determination by cryo-EM

Image stacks were × 2 binned in Fourier space, drift-corrected, dose-weighted, and summed with MotionCor2 (*Zheng et al., 2017*). Contrast transfer function (CTF) parameters were estimated with CTFFIND (*Rohou and Grigorieff, 2015*) and a few thousand particles were manually picked up with EMAN2 (*Tang et al., 2007*) for initial templates. Automatic particle picking and reference-free 2D classification were carried out with RELION (*Scheres, 2012*). This yielded clear hexameric and pentameric 2D class averages. An initial 3D structure of the hexamer was reconstructed from well-defined class averages at pH 8.0 through probabilistic orientation assignment by PRIME (*Elmlund et al., 2013*). GPU-accelerated RELION (*Kimanius et al., 2016*) was used for all the following steps unless noted otherwise. Particles belonging to the selected classes were refined against the initial structure by 3D auto refinement. This gave a ~ 8.5 Å-resolution map based on the gold standard Fourier shell correlation (FSC) criteria (*Chen et al., 2013*), where the FSC between two volumes, each independently generated from half the data set, drops to 0.143. The map resolved some α-helices running through the cytoplasmic, TM and periplasmic domains. The map was used to construct the initial 3D atomic model of the hexamer for solving the crystal structure as described.

For final reconstructions, initial 3D references were generated from the crystal structures of the pentamer (accession code: 5SV0; *Celia et al., 2016*) and the hexamer in this study using EMAN2 (*Tang et al., 2007*), and then low-pass filtered to 30 Å for the hexamer and 20 Å for the pentamer. Particles at pH 8.0 and 5.4 were subjected to 3D classification and 3D auto refinement. No symmetry was applied during these procedures. The numbers of particles for final 3D reconstruction of the hexameric complex was 38,323 from a total of 276,526 picked-up particles for datasets at pH 8.0. Reconstructions from dataset at pH 5.4 resolved a pentameric structure with clear α-helices in the cytoplasmic domain but poorer masses in the TM region except for a rod-like density inside the channel. To improve the structure, 5-fold symmetry was enforced for 3D classification and 3D auto refinement. The numbers of particles for final 3D reconstruction of the pentameric complex was 22,243 from 122,908 picked-up particles.

A soft mask was estimated and applied to the two half-maps in the post-processing process of RELION (*Scheres, 2012*). B-factor estimation and map sharpening were also carried out in the post-processing step. The resolutions based on the gold standard FSC 0.143 criteria (*Chen et al., 2013*) were measured to be 6.69 Å for the hexamer at pH 8.0 and 7.11 Å for the pentamer at pH 5.4. Local resolution was estimated using ResMap (*Kucukelbir et al., 2014*) and Blocres in the Bsoft package (*Heymann and Belnap, 2007*) from unfiltered half maps. Both programs gave similar estimates, but the latter values appeared to be of slightly lower resolution. Details related to cryo-EM SPA are summarized in *Supplementary file 1*-Table S2.

## Construction of the ExbB-ExbD model

The ExbD TM helix ($ExbD_{TM}$) was determined by hydrophobicity calculation along the sequence. Atomic models of the ExbB hexamer and $ExbD_{TM}$ trimer ($ExbB_6ExbD_{3TM}$) and the ExbB pentamer and $ExbD_{TM}$ monomer ($ExbB_5ExbD_{1TM}$) were constructed based on the cryo-EM maps of the hexameric and pentameric complexes, respectively, by using COOT (*Emsley et al., 2010*). Phe 23 and Pro22 in $ExbD_{TM}$ were adjusted to the TM level of aromatic residues such as Tyr 195, Tyr 132, Phe 198, Phe 41, Phe 42, and Trp 38 in ExbB. The detergent belt encloses these residues near the cytoplasmic surface (*Figure 6—figure supplement 1A*). Then, the models were fitted into the maps with rigid-body refinement of individual ExbB subunits and $ExbD_{TM}$ in real space using Phenix.refine (*Adams et al., 2010*). This gave good geometries and high cross correlation (CC) values against the cryo-EM maps for both the models (*Supplementary file 1*-Table S3). Structure figures in *Figures 1, 2,7A,B and D*, *Figure 1—figure supplement 2* and *Figure 2—figure supplement 1* were prepared

with PyMol (The PyMOL Molecular Graphics System, Schrödinger, LLC), and those in *Figure 6A–C* , *Figure 4—figure supplement 1D* and *Figure 6—figure supplement 1* were prepared with UCSF Chimera (*Pettersen et al., 2004*). Electrostatic distributions were calculated using PDB2PQR and displayed using APBS (*Unni et al., 2011*).

### Channel current recording

The lipid mixture of DOPG, DOPC and DOPE at a molar ratio 2:3:5 (*Celia et al., 2016*) was spread over a 200 µm diameter hole in a 0.5 mm thick Teflon sheet partition. A planar lipid bilayer formed at the hole separated two chambers filled with 2 mM KPi at pH 7.5. Purified ExbBD complexes were mixed with liposomes of the same lipid mixture at a lipid-protein weight ratio of 20:1 or 10:1. After 2 hr incubation at room temperature, the detergent was removed with a Bio-Spin six column (Bio-Rad) equilibrated with 10 mM KPi, pH 7.5, 50 mM KCl, and 0.3 M sucrose. Proteoliposomes were then injected from the cis chamber and fused into the bilayers. The reference electrode was placed in the cis chamber. Current data were recorded and stored in a PC using pCLAMP software (Molecular Devices, Sunnyvale, CA) through Axopatch 200B amplifier and Digidata 1550A digitizer (Molecular Devices). The low pass filter was set to 1 kHz for the cut-off frequency, and data were sampled at 10 kHz. To change pH, 0.5 M succinic acid, pH 4.0 or 0.5 M Tris-HCl, pH 8.95 was added to both chambers.

### Preparation, electron microscopy and image analysis of 2D crystals

ExbBD complexes purified with $C_8E_4$ were mixed with *E. coli* total lipid extract (Avanti Polar Lipids). The samples were dialyzed over a semipermeable membrane with a molecular weight cut-off of 10,000 to remove the detergent in 150 mM NaCl, 1 mM TCEP, 0.01% $NaN_3$, and 25 mM Na-citrate, pH 5.4 or 25 mM HEPES, pH 7.0 or 25 mM glycine, pH 9.0 at 4°C. The samples were negatively-stained with 2% (w/v) uranyl acetate and examined with the JEOL-2100 electron microscope (JEOL). Many 2D crystals were formed at pH 5.4, but few at pH 7.0 and none at pH 9.0. Crystal images showing diffraction spots were selected and analysed by Fourier filtering of diffraction spots with a modified version of the MRC package (*Crowther et al., 1996*). Crystals were centrifuged and the pellet was washed with the buffer. This process was repeated twice. Then, the sample was dissolved in SDS sample buffer, heated at ~ 95°C for 5 min, and run on a polyacrylamide gel.

## Acknowledgements

We thank M Watanabe for the early stage of this work, K Hirata, K Yamashita, Y Kawano, M Yamamoto, G Ueno, and other beam line staff for their help in X-ray diffraction data acquisition at SPring-8, MB Braunfeld, AF Brilot, KA Verba, DA Agard (UCSF), A Siebert, Y Chaban, C Hecksel (Diamond), H Shigematsu, T Yokoyama (RIKEN), K Mitsuoka (Osaka University), and S De Carlo (NeCEN) for cryo-EM data collection, and S Zheng for allowing us use of MotionCor2 before the software was available to the public. We are also grateful to T Suzuki and N Dohmae for peptide mass finger printing analysis, S Oiki for use of electrophysiological devices, and D B McIntosh for help in improving the manuscript. This work was supported by the Japan Society for the Promotion of Science Grant-in-Aid for Scientific Research Grant Number 15K06986 to (SY-M), 16H04757 (to KY), 15H04675 and 17H05876 (to HS). X-ray diffraction experiments were performed at BL32XU, BL41XU and BL26 in SPring-8 with the approval of the Japan Synchrotron Radiation Research Institute (JASRI) and RIKEN (Proposal Nos. 2012B1061, 2013B1085, 2014B1045, 2015A1021, 2015B2021, 2016A2530, and 2016B2530). We also acknowledge Diamond for access and support of the Cryo-EM facilities at the UK national electron bio-imaging centre (eBIC), proposal EM14838-1 and EM16817-1, funded by the Wellcome Trust, MRC and BBSRC.

## Additional information

### Funding

| Funder | Grant reference number | Author |
|---|---|---|
| Japan Society for the Promotion of Science | 15K06986 | Saori Maki-Yonekura |

| Japan Society for the Promotion of Science | 15H04675 | Hirofumi Shimizu |
| Japan Society for the Promotion of Science | 17H05876 | Hirofumi Shimizu |
| Japan Society for the Promotion of Science | 16H04757 | Koji Yonekura |
| Japan Society for the Promotion of Science | 20370064 | Koji Yonekura |

The funders had no role in study design, data collection and interpretation, or the decision to submit the work for publication.

## Author contributions

Saori Maki-Yonekura, Conceptualization, Data curation, Formal analysis, Supervision, Funding acquisition, Validation, Investigation, Methodology; Rei Matsuoka, Yoshiki Yamashita, Data curation, Formal analysis, Visualization; Hirofumi Shimizu, Data curation, Formal analysis, Funding acquisition, Validation, Investigation, Methodology; Maiko Tanaka, Fumie Iwabuki, Data curation, Investigation, Sample production, purification and crystallization; Koji Yonekura, Conceptualization, Resources, Data curation, Software, Formal analysis, Supervision, Funding acquisition, Validation, Investigation, Visualization, Methodology, Writing—original draft, Project administration, Writing—review and editing

## Author ORCIDs

Koji Yonekura https://orcid.org/0000-0001-5520-4391

## Decision letter and Author response

Decision letter https://doi.org/10.7554/eLife.35419.034
Author response https://doi.org/10.7554/eLife.35419.035

# Additional files

## Supplementary files

• Supplementary file 1. Table S1: Data collection and refinement statistics of the ExbB and ExbD crystals grown Table S2. Data collection and image analysis statistics for single particle cryo-EM. Table S3. Refinement statistics of atomic models against the cryo-EM maps
DOI: https://doi.org/10.7554/eLife.35419.017
• Transparent reporting form
DOI: https://doi.org/10.7554/eLife.35419.018

## Major datasets

The following datasets were generated:

| Author(s) | Year | Dataset title | Dataset URL | Database, license, and accessibility information |
|---|---|---|---|---|
| Maki-Yonekura S, Matsuoka R, Yonekura K | 2018 | Structure of the ExbB/ExbD hexameric complex | http://www.rcsb.org/pdb/search/structidSearch.do?structureId=5ZFP | Publicly available at the RSCB Protein Data Bank (accession no. 5ZFP) |
| Yonekura K, Yamashita Y, Matsuoka R, Maki-Yonekura S | 2018 | Structure of the ExbB/ExbD hexameric complex (ExbB6ExbD3TM) | http://www.rcsb.org/pdb/search/structidSearch.do?structureId=5ZFU | Publicly available at the RSCB Protein Data Bank (accession no. 5ZFU) |
| Yonekura K, Yamashita Y, Matsuoka R, Maki-Yonekura S | 2018 | Structure of the ExbB/ExbD hexameric complex (ExbB6ExbD3TM) | http://www.ebi.ac.uk/pdbe/entry/emdb/EMD-6927 | Publicly available at the Electron Microscopy Data Bank (accession no. 6927) |

| Yonekura K, Yamashita Y, Matsuoka R, Maki-Yonekura S | 2018 | Structure of the ExbB/ExbD pentameric complex (ExbB5ExbD1TM) | http://www.rcsb.org/pdb/search/structidSearch.do?structureId=5ZFV | Publicly available at the RSCB Protein Data Bank (accession no. 5ZFV) |
| Yonekura K, Yamashita Y, Matsuoka R, Maki-Yonekura S | 2018 | Structure of the ExbB/ExbD pentameric complex (ExbB5ExbD1TM) | http://www.ebi.ac.uk/pdbe/entry/emdb/EMD-6928 | Publicly available at the Electron Microscopy Data Bank (accession no. 6928) |

The following previously published datasets were used:

| Author(s) | Year | Dataset title | Dataset URL | Database, license, and accessibility information |
|---|---|---|---|---|
| Celia H, Noinaj N, Zakharov SD, Bordignon E, Botos I, Santamaria M, Barnard TJ, Cramer WA, Lloubes R, Buchanan SK | 2016 | Structure of the ExbB/ExbD complex from E. coli at pH 7.0 | https://www.rcsb.org/structure/5SV0 | Publicly available at the RCSB Protein Data Bank (accession no. 5SV0) |
| Celia H, Noinaj N, Zakharov SD, Bordignon E, Botos I, Santamaria M, Barnard TJ, Cramer WA, Lloubes R, Buchanan SK | 2016 | Structure of the ExbB/ExbD complex from E. coli at pH 4.5 | https://www.rcsb.org/structure/5sv1 | Publicly available at the RCSB Protein Data Bank (accession no. 5SV1) |

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
