## [Decision Letter]

[Editors’ note: a previous version of this study was rejected after peer review, but the authors submitted for reconsideration. The first decision letter after peer review is shown below.]

Thank you for submitting your work entitled "Hexameric and pentameric complexes of the ExbBD energizer in the Ton system" for consideration by *eLife*. Your article has been evaluated by a Senior Editor and three reviewers, one of whom is a member of our Board of Reviewing Editors. The following individuals involved in review of your submission have agreed to reveal their identity:, Nikolaus Grigorieff (Reviewer #1); Phillip Klebba (Reviewer #2).

Our decision has been reached after consultation between the reviewers. Based on these discussions and the individual reviews below, we regret to inform you that your work will not be considered further for publication in *eLife*.

The reviewers recognize that you present new and interesting structures of the ExbBD sub-complex, which is part of the Ton complex. However, the lack of clear mechanistic implications makes the manuscript unsuitable for publication in *eLife*. If, at a later stage, you have new data to tie the new structures to a mechanism, I encourage you to resubmit a revised manuscript.

Summary:

The authors present an interesting study of different oligomeric states of the bacterial Ton complex that is involved in transducing the energy of an electrochemical gradient across the inner membrane, where it resides, to drive transport across the outer membrane. It is currently unknown how the Ton complex harvests the proton gradient and how it connects to the outer membrane to transfer the harvested energy to transporters.

The authors looked at a sub-complex consisting of ExbB and ExbD subunits with stoichiometries that are pH-dependent. Previously, only a low-pH form was known at atomic detail consisting of five ExbB subunits and one ExbD subunit. The authors show that there is an equilibrium between this "pentameric" form and a "hexameric" form that includes six ExbB and three ExbD subunits. They confirm the co-existence of these two forms by electron crystallography of 2D crystals grown at a single (intermediate) pH showing different crystal types made of the two different oligomeric forms. The equilibrium between the two forms is shifted towards the hexameric form at higher pH. The authors also obtained an X-ray crystal structure of the ExbB hexamer, albeit with very little ExbD, and cryo-EM structures of the "pentameric" and "hexameric" forms at low and high pH, respectively. They correlate the pH-dependent shift in the equilibrium with changes in the ion conductance of the complex across membranes, measured by single-channel and multichannel measurements, and show that it is reversible. The Ton complex has pronounced selectivity for cations and conductance increases as the equilibrium shifts from the pentameric form to the hexameric form, presumably because the size of the pore formed by ExbB is larger in the hexamer compared to the pentamer. Finally, the presence of asymmetry in the ExbB hexamer is an interesting detail that is likely important for the mechanism of the Ton complex.

The description of the hexameric form and its dynamic equilibrium with the pentameric form is interesting and may represent a significant advance in the field. However, it remains unclear how the new structural insights advance our understanding of the molecular mechanism of the Ton complex. The authors offer some speculations on what the hexameric form and other new structural details mean for the energy transduction mechanism and coupling to transporters in the outer membrane. Unfortunately, these speculations are quite vague and the mechanistic insight gained from the structures presented here is very limited. Furthermore, the authors offer no explanation of how the dynamic conversion between pentameric and hexameric forms occurs, which in itself would be an interesting mechanism to study. It is unclear how the purified complex relates to the active complex in bacteria, especially since important parts of the machinery are missing or unresolved, including TonB and parts of ExbD. Presumably, TonB plays an important role in the transduction mechanism as it is thought to interact with the transporters in the outer membrane.

The lack of clear mechanistic implications is the main weakness of the manuscript, making it unsuitable for publication in *eLife*. If, at a later stage, the authors have new data to tie their new structures to a mechanism, they are encouraged to resubmit a revised manuscript. The full reviews are appended below for your reference.

*Reviewer #1:*

The authors investigated different oligomeric states of the bacterial Ton complex that is involved in transducing the energy of an electrochemical gradient across the inner membrane, where it resides, to drive transport across the outer membrane. It is currently unknown how the Ton complex harvests the proton gradient and how it connects to the outer membrane to transfer the harvested energy to transporters.

The authors looked at a sub-complex consisting of ExbB and ExbD subunits with stoichiometries that are pH-dependent. Previously, only a low-pH form was known at atomic detail, consisting of five ExbB subunits and one ExbD subunit. The authors show that there is an equilibrium between this "pentameric" form and a "hexameric" form that includes six ExbB and three ExbD subunits. They confirm the co-existence of these two forms by electron crystallography of 2D crystals grown at a single (intermediate) pH, showing different crystal types made of the two different oligomeric forms. The equilibrium between the two forms is shifted towards the hexameric form at higher pH. The authors also obtained an X-ray crystal structure of the ExbB hexamer, albeit with very little ExbD, and cryo-EM structures of the "pentameric" and "hexameric" forms at low and high pH, respectively. They correlate the pH-dependent shift in the equilibrium with changes in the ion conductance of the complex across membranes, measured by single-channel and multichannel measurements. The Ton complex has pronounced selectivity for cations and conductance increases as the equilibrium shifts from the pentameric form to the hexameric form, presumably because the size of the pore formed by ExbB is larger in the hexamer compared to the pentamer.

An interesting detail of the hexameric cryo-EM structure is that is does not display any symmetry: the subunits in the ExbB hexamer are somewhat staggered along the pseudo-symmetry axis and the ExbD trimer is off-center inside the ExbB hexamer. The authors offer some speculations on what this may mean for the energy transduction mechanism and coupling to transporters in the outer membrane. Unfortunately, these speculations are quite vague and the mechanistic insight gained from the structures presented here is very limited. The description of the hexameric form, a pseudo-atomic model of it, and an electrophysiological characterization of the different forms together may represent a significant advance in our understanding of these complexes. However, the authors must explain more clearly how these results help us understand the mechanism of the Ton complex.

*Reviewer #2:*

The authors report the architecture of a purified TonBExbBD complex that normally resides in the cytoplasmic membrane of *E. coli*. These proteins are proposed to transduce energy between the inner and outer membranes of the Gram-negative bacterial cell envelope, in a way that allows the active transport of iron by cell surface siderophore receptor proteins. In this manuscript the authors present crystallographic and other data suggesting the stoichiometry of the various proteins in the TonBExbBD complex. It is not the first effort to either quantify or structurally define the system. Buchanan and colleagues already did so in their 2016 paper (Celia, Nature 538: 60-65, 2016). The Yonekura manuscript describes the same proteins, with about the same findings. Consequently, this paper is more confirmatory than discovery, and in that way lacks the novelty one might expect for publication in *eLife*. Along those lines, Celia et al. did not report the 6:3 ExbB:D form: now the 4 or more groups that estimated the stoichiometry of purified, reconstituted TonBExbBD all reached different conclusions. Which of these are pertinent in vivo? The authors ultimately propose that the mechanism of TonBExbBD utilization of PMF involves conversion between pentameric/hexameric forms of the ExbB component, and monomeric/trimeric forms of the ExbD component: an ExbBExbD(2) subunit moves into and out of the complex in response to movement of protons. This is a novel idea. However, its feasibility is in doubt, and how might TonB fit into it? It's also possible, as the authors acknowledge, that one of the oligomeric complexes is a crystallization artifact. They do not sort this out with experiments, nor determine what actually occurs in living cells. They put a mechanistic spin on their pentameric-hexameric forms, but offer no evidence of a biologically relevant conversion between them. One thing that's problematic about the mechanistic conclusions is that the change in pH that's proposed to convert the complex between pentamer and hexamer (pH 4-5 vs. pH 8-9) likely never occurs in actively metabolizing, iron transporting bacterial cells: the pH near the inner bacterial membrane will remain acidic in actively growing cells.

The paper contains a large amount of technically acceptable research. The authors reveal the ExbB hexamer at 2.8 Å; the overall resolution of the ExbBD structures is only 6-9 Å. The manuscript reports new attributes of ExbBD from their X-ray data and cryoEM: the alkaline pH-driven hexameric ExbB component purportedly (i.e., no data) has an ExbD trimer at its core, rather than the monomer proposed in the pentamer crystal, that was seen in both their experiments and those of Celia et al. Hence, Yonekura et al. infer a base ExbB:ExbD stoichiometry of 6:3 under one condition, and, along with Celia et al., a complex 5:2 in another condition. The images of ExbB may provide some insight into potential mechanisms. But, even if protons (pH) drive a change in ExbBD organization, from hexamer at alkaline pH to pentamer at acidic pH, it's anyone's guess how this change in quaternary structure might deliver an energy transducing force to the outer membrane. The authors do not substantially explore this point. One of the weaknesses of this article is that it does not reconcile other data nor consider existing models on TonB-dependent transport (Chang et al., 2001; Klebba, 2016). Yet, what's truly missing from their results are any data on TonB (and ExbD) that reveal what is the relevant structure of the complex, in vitro or in vivo. Consequently, the paper only makes an incremental advance in its own narrow field (that may be artifactual) and does not have broader impact on the bioenergetics of Gram-negative bacterial active transport through the outer membrane, or membrane biochemistry in general. This lack of significance and scope makes me question its appropriateness in this venue, and its value to the overall understanding of TonB-dependent transport.

*Reviewer #3:*

In this manuscript, the high-resolution "hexameric" complex of ExbB and ExbD was reported for the first time by crystal and cryo-EM structural analyses. Authors showed that co-existence of pentameric and hexameric complexes, and demonstrated that the latter complex appeared exclusively at basic pH (9.0) and exhibited more channel conductance. The structure of hexameric complex revealed that six subunits of ExbB and three ExbD transmembrane helices enclosed within the central channel. Based on the structures and functional analyses, authors proposed the model for ExbB/D activation and usage of proton motive force. Although this is not the first report of the structure of ExbB/D complex, I think the findings reported in this manuscript, the new hexameric complex and its active properties, are novel and provide new insights into understanding of the functional mechanism based on the high-resolution structural information, especially on the activation/inactivation step in this machinery. Experiments are well conducted and presented, and interpretation is reasonable. However I feel some descriptions in the Discussion are speculative and more discussion for activation mechanism is required. Specific comments are listed below:

1) In the single channel recording of ExbB/D complex shown in Figure 3A, lower panel, two increments of channel current are shown (160pS then 60pS). Does this reflect opening of two channels in the same time (one in low conductance state and the other in high conductance)? Since in Figure 3B there are no results shown for 220pS, it is unlikely that this trace represents channel activation moment. Showing longer traces would be informative to understand the channel performance.

2) “The two states / channels appear interchangeable in the lipid bilayer as macroscopic currents are reversible on adjusting pH (Figures 3C and D)”, and” Thus, the channel activity might be controlled by joining / ejecting two ExbD molecules and one ExbB through partial disruption of the structures”, I agree with this interpretation. If so, sub-complex consisting of one molecule of ExbB and two molecules of ExbD would be detected at lower pH. Would it be possible to detect such sub-complex by the analytical size exclusion column experiment, simply purified hexameric complex were run at pH 9.0?

3) I think "rotation mechanism" proposed in the Discussion is interesting and plausible, but in this stage, it is speculative. The only supporting evidence derived from the data presented here is asymmetric helix disposition of ExbD, so I thought it may be better not to emphasize rotation model at this stage. On the other hand, authors did not discuss how the activation happens on the basis of their structure. The notable difference and new important finding from the previous study (Celia et al.) is the hexameric complex, which contains three helices within the channel, showed higher conductance. Celia et al. showed that ExbB pentamer, which does not have ExbD plug in the channel, showed high conductance. But here, the presence of ExbD in the channel still allows high ion conduction. How does this happen? Such point should be discussed. Also, as pointed in 2), Figure 3 showed that ExbB/D complex seems to dynamically changes its active and inactive state by incorporating/dissociating the ExbB1/ExbD2 sub-complex, and it is regulated by the pH. I would like to know authors' idea of how this happens.

4) In the second paragraph of the subsection “A model for utilization of proton motive force”, authors emphasized that the functional importance of ExbD Asp25. However, in Celia et al., ExbB/D complex with D25A mutation in ExbD did not change the channel conductance. I would like to know the authors' idea regarding the function of D25, from the structural point of view, from their hexameric complex.

[Editors’ note: what now follows is the decision letter after the authors submitted for further consideration.]

Thank you for submitting your article "Hexameric and pentameric complexes of the ExbBD energizer in the Ton system" for consideration by *eLife*. Your article has been favorably evaluated by Richard Aldrich (Senior Editor) and three reviewers, one of whom is a member of our Board of Reviewing Editors. The following individuals involved in review of your submission have agreed to reveal their identity: Nikolaus Grigorieff (Reviewer #1); Seiji Kojima (Reviewer #2).

The reviewers have discussed the reviews with one another and the Reviewing Editor has drafted this decision to help you prepare a revised submission.

Summary:

The authors submitted a revised manuscript that represents a significant improvement over the original manuscript. They have added a description of how a proton gradient may cause rotation of TonB that ultimately drives transport across the outer membrane. They also suggest a mechanism that drives interconversion between pentameric and hexameric forms of ExbB, based on the protonation states of side chains inside the channel. While these hypotheses remain untested, they suggest new experiments that will shed more light on the energy transduction mechanism. Overall, the manuscript reports a large amount of new data, new structures and interesting ideas of how this fascinating molecular machine may utilize a proton gradient to drive membrane transport. Clearly, more work will have to be done but the present manuscript represents a significant advance that warrants publication in *eLife*.

The following points must be addressed by the authors before the manuscript can be published:

1) Density maps and atomic coordinates of the models described in the manuscript must be deposited in the EMDB and PDB.

2) The authors should explain the loss of ExbD in the crystal structures.

3) The authors should provide a structural reason for the observed number of ExbD helices in ExbB pentamers and hexamers and explain why other numbers do not occur. They should consider the role of symmetry in the ExbB complex in their discussion.

*Reviewer #1:*

This is a resubmission of a manuscript originally submitted last November. The authors present an interesting study of different oligomeric states of the bacterial Ton complex that is involved in transducing the energy of an electrochemical gradient across the inner membrane, where it resides, to drive transport across the outer membrane. It is currently unknown how the Ton complex harvests the proton gradient and how it connects to the outer membrane to transfer the harvested energy to transporters.

The authors looked at a sub-complex consisting of ExbB and ExbD subunits with stoichiometries that are pH-dependent. Previously, only a low-pH form was known at atomic detail consisting of five ExbB subunits and one ExbD subunit. The authors show that there is an equilibrium between this "pentameric" form and a "hexameric" form that includes six ExbB and three ExbD subunits. They confirm the co-existence of these two forms by electron crystallography of 2D crystals grown at a single (intermediate) pH showing different crystal types made of the two different oligomeric forms. The equilibrium between the two forms is shifted towards the hexameric form at higher pH. The authors also obtained an X-ray crystal structure of the ExbB hexamer, albeit with very little ExbD, and cryo-EM structures of the "pentameric" and "hexameric" forms at low and high pH, respectively. They correlate the pH-dependent shift in the equilibrium with changes in the ion conductance of the complex across membranes, measured by single-channel and multichannel measurements, and show that it is reversible. The Ton complex has pronounced selectivity for cations and conductance increases as the equilibrium shifts from the pentameric form to the hexameric form, presumably because the size of the pore formed by ExbB is larger in the hexamer compared to the pentamer. Finally, the presence of asymmetry in the ExbB hexamer is an interesting detail that is likely important for the mechanism of the Ton complex.

The description of the hexameric form and its dynamic equilibrium with the pentameric form is interesting and may represent a significant advance in the field. In the original manuscript, it remained unclear how the new structural insights advance our understanding of the molecular mechanism of the Ton complex. The authors have now added a description of how a proton gradient may cause rotation of TonB that ultimately drives transport across the outer membrane. They also suggest a mechanism that drives interconversion between pentameric and hexameric forms of ExbB, based on the protonation states of side chains inside the channel. While these hypotheses remain untested, they suggest new experiments that will shed more light on the energy transduction mechanism.

The manuscript reports a large amount of new data, new structures and interesting ideas of how this molecular machine may utilize a proton gradient to drive membrane transport. Clearly, more work will have to be done but the present manuscript represents a significant advance that warrants publication in *eLife*.

*Reviewer #2:*

This revised version significantly improved the manuscript. The previous version lacks the mechanistic explanation of the function and activation of ExbB/D complex from the point of new structural features discovered by the authors. In my opinion, this time these points are explained clearly based on their findings. Last time I requested to discuss the reason why existence of three helices of ExbD allows the complex to higher conductance, and now authors show the reasonable interpretation (pore size) based on the calculation using the newly developed structural model. Authors also described their idea how the channel activation occurs. It is plausible that protonation state of the acidic residues in the cytoplasmic tip of the complex is key to induce transition between hexamer-pentamer complexes. Authors also include the TonB story in their model, and rotation of the transmembrane segment is now more understandable than previous version. Although the basic structure of the ExbB/D complex is already published by Celia et al., and some parts of their structural finding are confirmatory, but I think the discoveries of the ExbB6/ExbD3 complex and transition between hexamer-pentamer complex related to activation mechanism are important enough, at least to provide testable ideas for researchers in the field. Therefore, I now feel comfortable that this version is suitable for *eLife* publication standard.

*Reviewer #3:*

ExbBD is a fascinating energizer and the fact that functional structures are becoming available is very exciting. This paper describes X-ray structure of the hexameric ExbB complexes (unfortunately missing ExbD) from crystals that diffract to similar resolution (2.84 Å) compared to the recently published 2.6 Å resolution structure of the pentameric ExbB (Celia et al., 2016) and the cryoTEM structures of the ExbB hexamer and ExbD trimer complex. The crystallographic statistics in the table are good enough and the quality of the cryoTEM map is also good from the figures provided and has allowed to construct the helices of the model at 6.7 Å. Although, more work will be necessary to complete the functional mechanism in the complex, this paper describes interesting differences between their structural models (which contains the hexameric cryoTEM structures with ExbB molecules with different features at pH 8.0) and the pentameric X-ray structure at pH 4.5 (Celia et al., 2016) and is thus worth a publication. Since the pentameric structures at pH 4.5 and the hexameric structure at pH 8.0 likely correspond to different states, publication of this new structure and release of coordinates to compare these two structures of ExbBD energizer complexes would be greatly beneficial for the community.

From these structures and previous work, the authors also proposed the transition mechanism between the hexameric and pentameric ExbBD complexes. In particular, they confirmed the coexistence of the hexameric and pentameric complexes within lipid bilayers. Their proposed molecular mechanism that deprotonation of the acidic residues in the channel is a key for the complex formation to function as an energizer is also worth a publication.

The authors should address following points.

1) Authors should submit their coordinates and experimental data to the worldwide Protein Data Bank (wwPDB) prior to a submission of manuscript. In addition, the submission with the wwPDB validation report is highly recommended.

2) The authors described the decrease of proportion of ExbD during crystallization process. Why did this happen?

3) The discussion on the transition mechanism between the hexameric and pentameric ExbBD is still a bit vague. Authors mentioned that three helices of ExbD are enough and four is too much. Can the hexamer accommodate two TM helices of ExbD? Why two ExbD molecules are always recruited during the transition from 5mer to 6mer? Does the symmetrical arrangement help to form the ExbD trimer within the ExbB hexamer?

---

## [Author Response]

[Editors’ note: the author responses to the first round of peer review follow.]

[…] However, it remains unclear how the new structural insights advance our understanding of the molecular mechanism of the Ton complex. The authors offer some speculations on what the hexameric form and other new structural details mean for the energy transduction mechanism and coupling to transporters in the outer membrane. Unfortunately, these speculations are quite vague and the mechanistic insight gained from the structures presented here is very limited.

We had shown that the hexameric and pentameric complexes coexist in solution and within lipid membrane. The samples contained both full-length ExbB and ExbD (Figure 1—figure supplement 3B and Figure 5E (new figure)). These results support the existence of the biological relevance of the ExbB hexameric complex. We have added a new figure built from an atomic model of the ExbB hexamer and ExbD trimer, consisting of full side chains. The new model has revealed electrostatic details of the ion path (Figure 6) and a cation-selective filter made of Asp 25 in the ExbD TM helix at the right place. The ExbB hexamer and ExbD trimer complex has a ~ 5.5 Å pore in the thinnest diameter (Figure 6C), which is well fitted with the reported pore size of cation channels. In contrast, the pore in the pentameric complex with a central helix is too small for permeation of cations (Figure 6C), and the structure of the pentameric complex would reduce the efficiency of energy transduction. Thus, the hexameric complex is most likely the active form. The manuscript also includes experiments on TonB-ExbB-ExbD complexes and the disposition of TonB on the structures, not inside but outside the ExbB ring. Due to this disposition, TonB would little affect complexation of ExbB and ExbD. This was also shown by Celia et al., 2016. In addition, comparison of the ExbB pentamer and hexamer structures suggests that protonation / deprotonation of acidic residues at the cytoplasmic end of the channel controls the interconversion of pentamer / hexamer (Figures 2B, 6D and 7B). Based on these findings, we have proposed improved models for utilization of proton motive force and oligomeric transformation (text and Figure 7).

Furthermore, the authors offer no explanation of how the dynamic conversion between pentameric and hexameric forms occurs, which in itself would be an interesting mechanism to study.

We added the explanation of our model on the conversion mechanism as below and in Figure 7.

“The equilibrium of hexamer and pentamer may depend on the protonation state of the acidic residues such as Glu 105, Glu 109, Asp 103 and so on at the cytoplasmic end of the channel (Figures 2B and 6D), in accordance with PMF. […] The channel in the hexameric complex can accommodate three TM helices but cannot four helices”.

It is unclear how the purified complex relates to the active complex in bacteria, especially since important parts of the machinery are missing or unresolved, including TonB and parts of ExbD. Presumably, TonB plays an important role in the transduction mechanism as it is thought to interact with the transporters in the outer membrane.

The samples we examined contained full-length ExbB and full-length ExbD (Figure 1—figure supplement 3B and Figure 6—figure supplement 1B). We had described that TonB likely resides around the ExbB complex, as the TonB TM residues interact with the first TM helix (α2) of ExbB that faces the lipid interface around the ExbB hexamer (Figures 1 and 5C). We have also tried to obtain stable complexes of TonB-ExbB-ExbD, TonB-ExbB and TonB-ExbD complexes for structural study but not succeed yet. The difficulties probably reflect that TonB tends to more dissociate from the complex due to this disposition, not inside but outside the ExbB ring. Thus, TonB would little affect complexation of ExbB and ExbD. This was also shown by Celia et al., 2016. We added this explanation in Discussion.

The lack of clear mechanistic implications is the main weakness of the manuscript, making it unsuitable for publication in eLife. If, at a later stage, the authors have new data to tie their new structures to a mechanism, they are encouraged to resubmit a revised manuscript. The full reviews are appended below for your reference.Reviewer #1:[…] An interesting detail of the hexameric cryo-EM structure is that is does not display any symmetry: the subunits in the ExbB hexamer are somewhat staggered along the pseudo-symmetry axis and the ExbD trimer is off-center inside the ExbB hexamer. The authors offer some speculations on what this may mean for the energy transduction mechanism and coupling to transporters in the outer membrane. Unfortunately, these speculations are quite vague and the mechanistic insight gained from the structures presented here is very limited. The description of the hexameric form, a pseudo-atomic model of it, and an electrophysiological characterization of the different forms together may represent a significant advance in our understanding of these complexes. However, the authors must explain more clearly how these results help us understand the mechanism of the Ton complex.

We have improved the manuscript and described mechanisms based on the structures. Please also refer to responses to the Summary above.

Reviewer #2:The authors report the architecture of a purified TonBExbBD complex that normally resides in the cytoplasmic membrane of E. coli. These proteins are proposed to transduce energy between the inner and outer membranes of the Gram-negative bacterial cell envelope, in a way that allows the active transport of iron by cell surface siderophore receptor proteins. In this manuscript the authors present crystallographic and other data suggesting the stoichiometry of the various proteins in the TonBExbBD complex. It is not the first effort to either quantify or structurally define the system. Buchanan and colleagues already did so in their 2016 paper (Celia, Nature 538: 60-65, 2016). The Yonekura manuscript describes the same proteins, with about the same findings. Consequently, this paper is more confirmatory than discovery, and in that way lacks the novelty one might expect for publication in eLife.

We disagree. In this report, we have shown a complex structure made of six ExbB and three ExbD molecules revealed by using X-ray crystallography and single particle cryo-EM. This complex has not been described before and the structure reveals a striking spiral arrangement of the six ExbB subunits and asymmetrical disposition of the three ExbD TM helices, both features likely key for channel function. The manuscript has also shown that hexameric and pentameric complexes coexist, with the proportion of hexamer increasing with pH. Channel current measurement and 2D crystallography all support the formation of the two oligomeric states in membrane. Based on our findings, we have proposed a model for activation / inactivation associated with hexamer and pentamer formation and utilization of proton motive force. We believe that these data and models provide significant advances in our understanding this complex system. Indeed, the other two reviewers acknowledged the novelty of this work. Please also refer to responses to the Summary above.

Along those lines, Celia et al. did not report the 6:3 ExbB:D form: now the 4 or more groups that estimated the stoichiometry of purified, reconstituted TonBExbBD all reached different conclusions. Which of these are pertinent in vivo?

We have shown that hexameric and pentameric complexes coexist both in detergent micelles and within membrane and the ratio of the pentameric and hexameric complexes changes depending on conditions (Figure 4). In an equilibrium of pentamers and hexamers, it is likely that interconversion occurs through the presence of monomers and multimers. This should explain why the number of ExbB and ExbD subunits in the functional unit has been argued for years (Higgs et al., 2002; Jana et al., 2011; Pramanik et al., 2011; Sverzhinsky et al., 2014; Sverzhinsky et al., 2015). A previous MS analysis showed that the main fraction of the sample in the weak basic condition used corresponded to ExbB hexamers and small amounts of the pentamer were also found (Pramanik et al., 2011). This proportion is consistent with our results. Other studies were based on rather crude methods such as cross-linking and negative-staining EM. We have also showed that 2D crystals contained both full-length ExbB and ExbD (Figure 5E). Please also see our fourth response to reviewer #2.

The authors ultimately propose that the mechanism of TonBExbBD utilization of PMF involves conversion between pentameric/hexameric forms of the ExbB component, and monomeric/trimeric forms of the ExbD component: an ExbBExbD(2) subunit moves into and out of the complex in response to movement of protons. This is a novel idea. However, its feasibility is in doubt, and how might TonB fit into it?

Please refer to our second and third responses to the Summary above.

It's also possible, as the authors acknowledge, that one of the oligomeric complexes is a crystallization artifact. They do not sort this out with experiments, nor determine what actually occurs in living cells.

Unlike the other examples such as MscL, V_0_, F_0_-ATPases and light harvesting complexes, we have observed the formation of the hexamer and pentamer in solution and within lipid bilayers and the distinct ion conductivities in the planar lipid membrane associated with pHs for the same ExbBD sample. These behaviors should reflect biological natures in vivo as described in Discussion. The samples in solution and 2D crystals contained both full-length ExbB and ExbD (Figure 1—figure supplement 3B and Figure 5E (new figure)). Thus, the hexameric and pentameric complexes are not a crystallization artifact. TonB would little affect complexation of ExbB and ExbD as described in our third response to the Summary above.

They put a mechanistic spin on their pentameric-hexameric forms, but offer no evidence of a biologically relevant conversion between them. One thing that's problematic about the mechanistic conclusions is that the change in pH that's proposed to convert the complex between pentamer and hexamer (pH 4-5 vs. pH 8-9) likely never occurs in actively metabolizing, iron transporting bacterial cells: the pH near the inner bacterial membrane will remain acidic in actively growing cells.

We have shown that the proportion of the hexameric and pentameric complexes depends on pH (Figure 4 and Table S2 in Supplementary file 1). The hexamer and pentamer were also observed at the neutral pH (Figure 4C and Table S2 in Supplementary file 1). The two states / channels appear interchangeable in the lipid bilayer as macroscopic currents are reversible on adjusting pH (Figures 3C and D). Combined with other data, these behaviors should reflect biological natures. We think that a local concentration of the hydronium around the cytoplasmic domains is critical, but the global pH near the inner membrane is not. The conversion mechanism was described in our second response to the Summary above.

The paper contains a large amount of technically acceptable research. The authors reveal the ExbB hexamer at 2.8 Å; the overall resolution of the ExbBD structures is only 6-9 Å. The manuscript reports new attributes of ExbBD from their X-ray data and cryoEM: the alkaline pH-driven hexameric ExbB component purportedly (i.e., no data) has an ExbD trimer at its core, rather than the monomer proposed in the pentamer crystal, that was seen in both their experiments and those of Celia et al. Hence, Yonekura et al. infer a base ExbB:ExbD stoichiometry of 6:3 under one condition, and, along with Celia et al., a complex 5:2 in another condition. The images of ExbB may provide some insight into potential mechanisms. But, even if protons (pH) drive a change in ExbBD organization, from hexamer at alkaline pH to pentamer at acidic pH, it's anyone's guess how this change in quaternary structure might deliver an energy transducing force to the outer membrane. The authors do not substantially explore this point. One of the weaknesses of this article is that it does not reconcile other data nor consider existing models on TonB-dependent transport (Chang et al., 2001; Klebba, 2016).

We have improved the explanation of the working mechanisms by reconciling the exiting models as below and in Figure 7. We also cited Chang, et al., 2001 and Klebba, 2016.

“The proton gradient through the spiral of ExbB subunits would generate a torsion force (Chang et al., 2001) and may induce rotational movement of the ExbD TM helices. […] This movement would facilitate to find the outer membrane receptors (Klebba, 2016) and interaction of the TonB C-terminal domain with the periplasmic domain of the receptor induces conformational changes to release a substrate bound to the receptor into the periplasmic space (Chang et al., 2001; Klebba, 2016)”.

Yet, what's truly missing from their results are any data on TonB (and ExbD) that reveal what is the relevant structure of the complex, in vitro or in vivo.

Please refer to our third response to the Summary above.

Consequently, the paper only makes an incremental advance in its own narrow field (that may be artifactual) and does not have broader impact on the bioenergetics of Gram-negative bacterial active transport through the outer membrane, or membrane biochemistry in general. This lack of significance and scope makes me question its appropriateness in this venue, and its value to the overall understanding of TonB-dependent transport.

Please refer to our first response to reviewer #2.

Reviewer #3:[…] However I feel some descriptions in the Discussion are speculative and more discussion for activation mechanism is required. Specific comments are listed below.

We have improved the manuscript by incorporating a new figure based on an atomic model of the ExbB hexamer and ExbD trimer (Figure 6), new crystallographic data (Table S1 in Supplementary file 1), experiments on TonB complexes (text). We have also proposed improved models for utilization of proton motive force and oligomeric transformation (text and Figure 7).

1) In the single channel recording of ExbB/D complex shown in Figure 3A, lower panel, two increments of channel current are shown (160pS then 60pS). Does this reflect opening of two channels in the same time (one in low conductance state and the other in high conductance)? Since in Figure 3B there are no results shown for 220pS, it is unlikely that this trace represents channel activation moment. Showing longer traces would be informative to understand the channel performance.

Yes. The data showed opening of two channels. Only incremental points as in Figure 3A were treated as the event of the single channel, since it is unlikely that two events occur simultaneously. This is the standard treatment for measurement of single channel conductance. We added this description in the legend of Figure 3. No single event jumping to ~ 220 pS was observed and longer traces only showed similar patterns as in Figure 3A.

2) “The two states / channels appear interchangeable in the lipid bilayer as macroscopic currents are reversible on adjusting pH (Figures 3C and D)”, and” Thus, the channel activity might be controlled by joining / ejecting two ExbD molecules and one ExbB through partial disruption of the structures”, I agree with this interpretation. If so, sub-complex consisting of one molecule of ExbB and two molecules of ExbD would be detected at lower pH. Would it be possible to detect such sub-complex by the analytical size exclusion column experiment, simply purified hexameric complex were run at pH 9.0?

We observed a small peak consisting of ExbB and ExbD at a lower molecular weight in the SEC patterns. The corresponding peak became smaller when run at pH 8.0 and disappeared at pH 9.0, which is consistent with the equilibrium of pentamer / hexamer. We added this description in text.

3) I think "rotation mechanism" proposed in the Discussion is interesting and plausible, but in this stage, it is speculative. The only supporting evidence derived from the data presented here is asymmetric helix disposition of ExbD, so I thought it may be better not to emphasize rotation model at this stage.

We have modified text not to emphasize the rotation. Please also refer to our sixth response to reviewer #2.

On the other hand, authors did not discuss how the activation happens on the basis of their structure. The notable difference and new important finding from the previous study (Celia et al.) is the hexameric complex, which contains three helices within the channel, showed higher conductance. Celia et al. showed that ExbB pentamer, which does not have ExbD plug in the channel, showed high conductance. But here, the presence of ExbD in the channel still allows high ion conduction. How does this happen? Such point should be discussed.

We have described the activation mechanism in the revised manuscript. Please refer to our sixth response to reviewer #3 below. The very high conductance (~ 220 pS) in Celia et al. (2016) was only observed for ExbB without ExbD. The behaviour of the channel with ExbD in our measurement is similar to that reported in Celia et al. (2016).

Also, as pointed in 2), Figure 3 showed that ExbB/D complex seems to dynamically changes its active and inactive state by incorporating/dissociating the ExbB1/ExbD2 sub-complex, and it is regulated by the pH. I would like to know authors' idea of how this happens.

The equilibrium of hexamer and pentamer may depend on the protonation state of the acidic residues such as Glu 105, Glu 109, Asp 103, and so on at the cytoplasmic end of the channel (Figures 2B and 6D), in accordance with PMF. If the hydronium concentration is elevated around this region, excess protonation of these residues might shift the equilibrium to denser packing among ExbB subunits by easing the electronegativity, which repels one ExbB subunit from the hexamer ring and excludes two ExbD from the central space to form the pentamer ring. Reversely, deprotonation of the acidic residues would shift to looser packing, which recruits two ExbD inside the ring and one ExbB to form the hexamer ring, if the local hydronium concentration is decreased. The channel in the hexameric complex can accommodate three TM helices but cannot four helices. We added this explanation in Discussion and Figure 7.

4) In the second paragraph of the subsection “A model for utilization of proton motive force”, authors emphasized that the functional importance of ExbD Asp25. However, in Celia et al., ExbB/D complex with D25A mutation in ExbD did not change the channel conductance. I would like to know the authors' idea regarding the function of D25, from the structural point of view, from their hexameric complex.

Asp 25 forms a cation selective filter, but the mutation of this aspartate does not affect the channel conductance (Celia et al., 2016). In our model, the aspartates reside at the lower part of the TM region and two of the aspartates facing to the pore produce an electronegative field inside the ion path (Figure 6B). The aspartates could allow only cations to pass through. We added this description in Discussion.

[Editors' note: the author responses to the re-review follow.]

The following points must be addressed by the authors before the manuscript can be published:1) Density maps and atomic coordinates of the models described in the manuscript must be deposited in the EMDB and PDB.

Done. Atomic coordinates and structure factors for the crystal structure of the ExbB hexamer and coordinates of ExbB_6_ExbD_3TM_ and ExbB_5_ExbD_1TM_ have been deposited in the Protein Data Bank under accession number 5ZFP, 5ZFU and 5ZFV respectively. Cryo-EM density maps of the hexameric and pentameric complexes have been deposited in the Electron Microscopy Data Bank under the accession number EMD-6927 and EMD-6928, respectively.

2) The authors should explain the loss of ExbD in the crystal structures.

The 2D crystals contained both full-length ExbB and ExbD (Figure 5E). The 3D crystals appeared to be built by stacking of 2D crystal layers (Figure 1—figure supplement 2A and B). The periplasmic part of ExbD, which includes a long loop and a compact peptidoglycan-binding motif, would be difficult to fit between the 2D crystal layers in both P21 and P1 3D crystal lattices without deforming these domains (Figure 1—figure supplement 2A and B). Thus, stacking of 2D crystal layers might exclude ExbD because of these steric effects. We added this explanation in text.

3) The authors should provide a structural reason for the observed number of ExbD helices in ExbB pentamers and hexamers and explain why other numbers do not occur. They should consider the role of symmetry in the ExbB complex in their discussion.

Two TM helices in the hexamer channel as well as no helix in the pentamer channel produce larger pores and would result in unproductive and high proton influx. Thus, ejecting three or one helices from the hexamer ring would be undesirable and the ExbD dimer, which is reported to be a stable complex, would be secreted / incorporated during the oligomeric transition from / to hexamer. This explanation is now in the text.

Reviewer #3:3) The discussion on the transition mechanism between the hexameric and pentameric ExbBD is still a bit vague. Authors mentioned that three helices of ExbD are enough and four is too much. Can the hexamer accommodate two TM helices of ExbD? Why two ExbD molecules are always recruited during the transition from 5mer to 6mer? Does the symmetrical arrangement help to form the ExbD trimer within the ExbB hexamer?

The ExbD TM trimer is asymmetrically located off-centre of the hexamer channel. We can’t see at the moment how the symmetrical arrangement of the six ExbB subunits could especially facilitate the formation of the ExbD trimer, except in so far as the hexamer ring provides the right size to accommodate the ExbD trimer and the right pore size for utilization of PMF.